# Triplet-driven chemical reactivity of β-carotene and its biological implications

Mateusz Zbyradowski [1], Mariusz Duda[1], Anna Wisniewska-Becker[1], Heriyanto [1,2], Weronika Rajwa[1], Joanna Fiedor [3], Dragan Cvetkovic [4], Mariusz Pilch[1,5] & Leszek Fiedor [1✉]

The endoperoxides of β-carotene (βCar-EPOs) are regarded as main products of the chemical deactivation of $^1O_2$ by β-carotene, one of the most important antioxidants, following a concerted singlet-singlet reaction. Here we challenge this view by showing that βCar-EPOs are formed in the absence of $^1O_2$ in a non-concerted triplet-triplet reaction: $^3O_2 + ^3$β-carotene → βCar-EPOs, in which $^3$β-carotene manifests a strong biradical character. Thus, the reactivity of β-carotene towards oxygen is governed by its excited triplet state. βCar-EPOs, while being stable in the dark, are photochemically labile, and are a rare example of nonaromatic endoperoxides that release $^1O_2$, again not in a concerted reaction. Their light-induced breakdown triggers an avalanche of free radicals, which accounts for the pro-oxidant activity of β-carotene and the puzzling swap from its anti- to pro-oxidant features. Furthermore, we show that βCar-EPOs, and carotenoids in general, weakly sensitize $^1O_2$. These findings underlie the key role of the triplet state in determining the chemical and photophysical features of β-carotene. They shake up the prevailing models of carotenoid photophysics, the anti-oxidant functioning of β-carotene, and the role of $^1O_2$ in chemical signaling in biological photosynthetic systems. βCar-EPOs and their degradation products are not markers of $^1O_2$ and oxidative stress but of the overproduction of extremely hazardous chlorophyll triplets in photosystems. Hence, the chemical signaling of overexcitation of the photosynthetic apparatus is based on a $^3$chlorophyll-$^3$β-carotene relay, rather than on extremely short-lived $^1O_2$.

[1] Faculty of Biochemistry, Biophysics and Biotechnology, Jagiellonian University, Gronostajowa 7, 30-387 Cracow, Poland. [2] Ma Chung Research Center for Photosynthetic Pigments, Ma Chung University, Villa Puncak Tidar N-01, Malang 65151, Indonesia. [3] Faculty of Physics and Applied Computer Science, AGH-University of Science and Technology, Mickiewicza 30, 30-059 Cracow, Poland. [4] Faculty of Technology, University of Niš, 16000 Leskovac, Serbia. [5] Faculty of Chemistry, Jagiellonian University, Gronostajowa 2, 30-387 Kraków, Poland. ✉email: leszek.fiedor@uj.edu.pl

Endoperoxides (EPOs), or cyclic dialkylperoxides, are a unique class of compounds whose properties are largely determined by a remarkable structural feature – the -O-O- bridge. Aromatic EPOs have attracted much attention, mainly as a convenient source of $^1O_2$ for a variety of applications, ranging from chemical synthesis to photomedicine[1]. This interest has been sparked both by the recognition of $^1O_2$ as a selective oxygenation agent and due to its strong cytotoxicity, making it one of the most dangerous reactive oxygen species (ROS) in biological systems[2–5]. The reactivity of $^1O_2$, as compared to the ground state molecular oxygen, $^3O_2$, is greatly enhanced due to the excess energy and the spin-allowed character of the reactions with other molecules that display singlet multiplicity, thus making $^1O_2$ one of the strongest known oxidizing species. Fortunately, $^3O_2$ remains inert towards organic compounds in their singlet states, owing to the resonance stabilization of its π-electron system[6] and Wigner's spin conservation rule. This permits life on our planet to exist under aerobic conditions at ambient temperatures.

Thermal, photochemical, or chemical activation of aromatic EPOs often leads to the quantitative release of $^1O_2$, and the mechanisms of attachment and release of $O_2$ have been thoroughly investigated, both experimentally and theoretically[7–11]. In most cases, the -O-O- bridge is formed in a concerted $[2+4]$ cycloaddition of a $^1O_2$ dienophile to a flat aromatic substrate[3,7,12]. Inversely, cycloreversion leads to the release of $^1O_2$ and the regeneration of hydrocarbon[8,9,11]. Another way the -O-O- bridge can be formed is the trapping of $^3O_2$, which is applied, e.g., in the synthesis of cyclic peroxides[13]. It is also an established experimental approach in the studies of biradicals[14,15].

So far, non-aromatic EPOs remain somewhat less elaborated, and their chemistry is not understood that well, perhaps with the exception of some naturally occurring ones, such as prostaglandin $G_2$, artemisinin, and the endoperoxides of β-carotene (βCar-EPOs, Fig. 1a)[9,16,17]. Our previous studies revealed that in a sensitized process all-trans-β,β-carotene (βCar) sequentially accumulates up to eight oxygen atoms, while its $C_{40}$-skeleton remains intact, yielding a series of βCar-EPOs[18], whose major products are β-carotene-5,8,-endoperoxide (βCar-5,8-EPO) and β-carotene-7,10-endoperoxide (βCar-7,10-EPO). To account for the pigment oxygenation mechanism, we proposed a concerted $[2+4]$ cycloaddition of $^1O_2$ to βCar in an s-cis-diene conformation, according to generally accepted mechanism of attachment of $^1O_2$ to dienes. The products of this chemical quenching appear to be responsible for the controversial pro-oxidant features of carotenoids (Crts), in particular βCar[19–21]. βCar-EPOs, found to rapidly accumulate in plants under high-light stress, were assumed to be the products of the in situ chemical quenching of $^1O_2$ by βCar[22]. Their accumulation, in particular of βCar-5,8-EPO, is correlated with chronic Crts oxygenation, the extent of photosystem II photoinhibition, and the expression of various $^1O_2$ marker genes[23]. The lower mass and volatile degradation products of βCar-EPOs, such as β-cyclocitral, play a crucial role in the chemical signaling of oxidative stress in oxygenic photosynthesis[23–25].

The chemical reactivity of Crts has to be viewed as one of the broad range of crucial roles that these simple isoprenoid pigments play in living organisms, including humans but, in particular, in photosynthetic organisms. Crts are considered to be the first line of defense against ROS, owing to their anti-oxidant features[26–29], and their capacity to nearly "catalytically" scavenge $^1O_2$ and chlorophyll (Chl) triplets through physical quenching[30–32]. The versatile (photo)protective functioning of Crts relies on their low-energy $T_1$ state, matching that of the $^1\Delta_g$ state. Crt $T_1$ is able to intercept the excitation energy from Chl triplets and $^1O_2$, and then to dissipate it harmlessly into the environment[5]. Practically each collision of Crt molecules with these excited species leads to

spin-allowed intramolecular excitation energy transfer (EET), and they are quenched almost within the diffusion limits[30]. The quenching of Chl triplets by Crts is especially important in photosynthetic pigment-proteins, in which Crts also play accessory antenna functions[33–36]. Recently, it has been found that Crt triplets can be efficiently generated via singlet fission in Crt aggregates or in photosynthetic antenna complexes, which is relevant to the quest to elevate the quantum performance of organic solar devices[37,38]. The efficiency of photosynthesis organisms and of such devices critically depends on (photo) protection, which ensures their durability under dense photon fluxes. In the photosynthetic apparatus, a major risk is associated with the excellent $^1O_2$ sensitizing properties of Chls, since their excited state relaxation is dominated by the $S_1$-$T_1$ intersystem crossing (ISC), despite the absence of heavy atoms in these molecules[39,40]. Paradoxically, photosynthetic solar energy conversion, the major source of bioenergy on the planet, relies entirely on singlet-singlet energy transfer, and useful photosynthetic reactions must necessarily compete with ISC in Chls. This latter process is wasteful in terms of energy and creates a severe risk of $^1O_2$ sensitization and oxidative damage to the photosynthetic machinery, the very origin of molecular oxygen.

Considering the mechanism of βCar-EPOs formation, the cycloaddition of $^1O_2$ to a 1,3-diene system with 6-s-cis conformation, present near the ionone rings in the βCar molecule[41], apparently may result in βCar-5,8-EPO. However, oxygenation to βCar-7,10-EPO requires an s-trans to s-cis conversion at the single $C_8$–$C_9$ bond, implying a different mechanism for the addition of $O_2$. Furthermore, chemically generated $^1O_2$ does not cause oxygenation, but only isomerization of βCar[42], whereas the lifetime and concentration of $^1O_2$ in our preparative system are expected to be strongly suppressed by both βCar and βCar-EPOs. Such pilling-up inconsistencies motivated us in this study to investigate in detail the mechanisms of the formation and breakdown of βCar-EPOs, and the involvement of $^1O_2$ and other ROS in these processes. We synthesized and characterized a series of βCar-EPOs, and their breakdown was monitored using HPLC and LC-MS/MS techniques and electron paramagnetic resonance (EPR), electronic absorption and time-resolved emission spectroscopies. In particular, our principal approach was to tune, in our preparative system, the level of $O_2$ and the lifetime of $^1O_2$ from the nanosecond to the millisecond range. In parallel, ab initio computations were performed to gain structural and thermodynamic insights. Here, we demonstrate that $^1O_2$ does not participate in oxygenation of βCar and the formation and breakdown of βCar-EPOs are not concerted reactions. Our study reveals a strong biradical character of the $T_1$ state as well as its key role in the photophysics and (photo)chemistry of Crts. βCar-EPOs are an uncommon example of non-aromatic EPOs that release $^1O_2$. We also show, for the first time, that Crts in solution weakly sensitize $^1O_2$.

## Results and discussion

**Synthesis and characterization of β-carotene endoperoxides.** A series of βCar-EPOs was photocatalytically synthesized from βCar following a previously described method[17], using red light (cut off >600 nm) and bacteriopheophytin a ($S_1$ maximum at 750 nm) as a photocatalyst (PC). Our previous study showed that acetone as the reaction milieu is optimal for obtaining various βCar-EPOs[17,18]. The most abundant ones, βCar-5,8-EPO and βCar-7,10-EPO, β-carotene-5,8,5′,8′-diendoperoxide (βCar-5,8,5′,8′-diEPO) and β-carotene-5,8,7′,10′-diendoperoxide (βCar-5,8,7′,10′-diEPO), shown in Fig. 1a, were isolated as before, using the RP-HPLC technique[17,18]. The shifts in their electronic spectra (Fig. 1a) reflect a gradual shrinkage of the π-electron

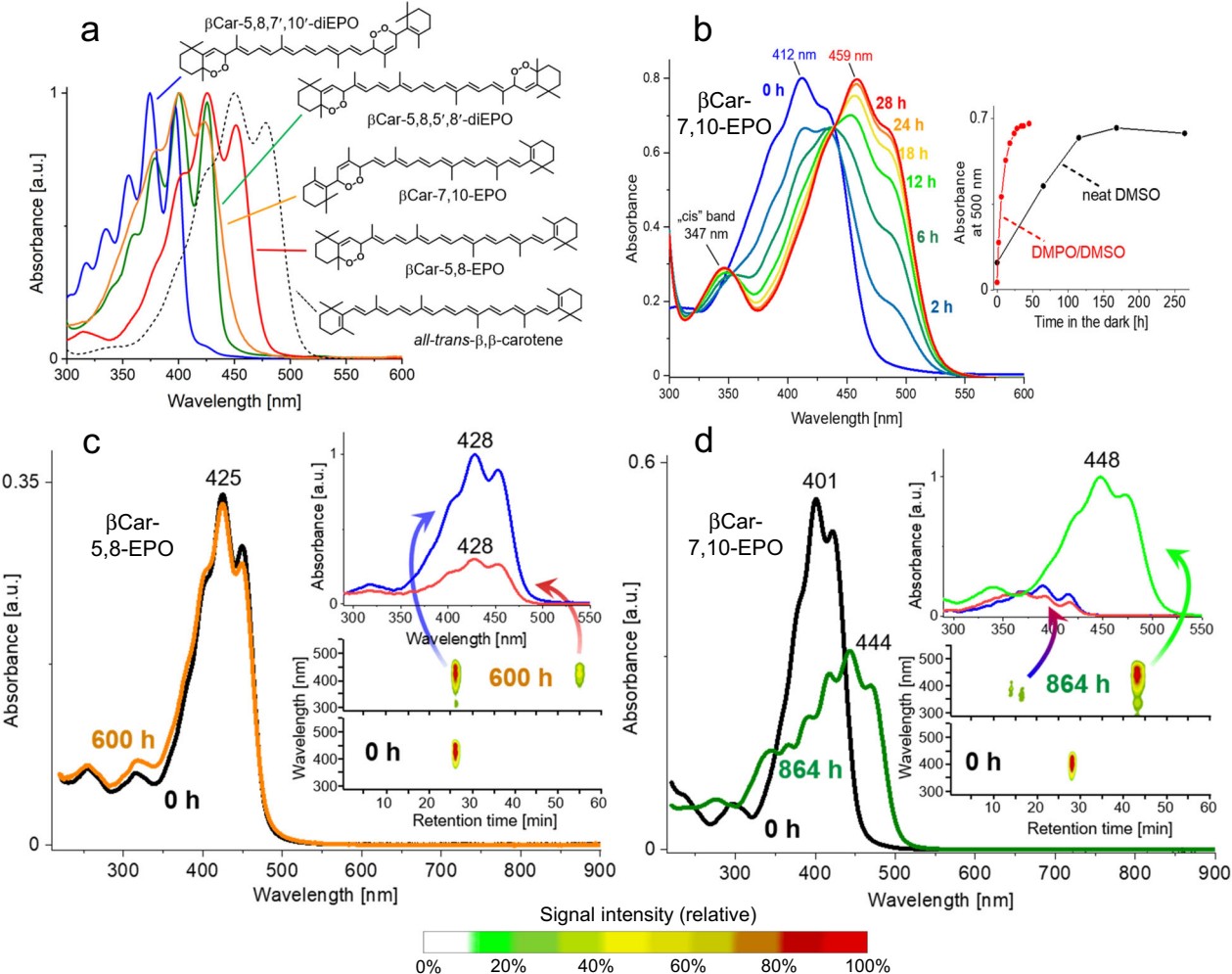

**Fig. 1 The structures and the electronic absorption spectra of β-carotene and its oxygenation products, and their breakdown in the dark. a** The structural formulae and electronic absorption spectra of β-carotene and β-carotene endoperoxides recorded in acetone at room temperature. **b** The spectral changes and kinetics (inset) of βCar-7,10-EPO breakdown in dimethyl sulfoxide (DMSO) in the presence of 5,5-dimethyl-1-pyrroline N-oxide (DMPO) as spin trap, in the dark at room temperature, monitored by electronic absorption spectroscopy. Note a 5-fold increase in the reaction rate caused by the addition of DMPO at 150 mM. **c, d** The breakdown of βCar-5,8-EPO (**c**) and βCar-7,10-EPO (**d**) kept in methanol in the dark and monitored by electronic absorption spectroscopy and HPLC with in-line absorption detection (insets). The arrows indicate the absorption spectra of the respective fractions. Source data are provided as a Source Data file.

chromophore system due to the attachment of oxygen molecules, from 11 conjugated C=C bonds in the parental βCar to 9, 7, and eventually to 6 conjugated C=C bonds in βCar-5,8,7′,10′-diEPO. The identity of the products was confirmed by detecting their molecular ions of m/z values near 569 and 601 in their mass spectra (Supplementary Figs. 1–4), which correspond to the presence of one or two -O-O- bridges, respectively. The results of the ab initio calculations on βCar and its two main oxygenation products confirm these assignments (Supplementary Tables 1 and 2). They also show a very weak solvent effect on the ground state, the $S_2$ state, and the orbital energies (HOMO and LUMO) of βCar and βCar-EPOs molecules (Supplementary Table 3). The stability of βCar-7,10-EPO is lower than that of βCar-5,8-EPO by 45–47 kJ/mol and does not depend on the medium. A slightly stronger solvent influence is seen in both the dipole moments and polarizabilities of the pigments. As expected, these parameters, with respect to βCar (plain hydrocarbon), increase in βCar-EPOs. The dipole moment of βCar-7,10-EPO is higher and its polariz-ability is evidently lower than that of βCar-5,8-EPO, which agrees well with the shorter conjugation length in the former EPO

($n=8$). This effect is also manifested in the lower value of both the LUMO-HOMO energy difference and the $S_2$ energy predicted for βCar-7,10-EPO. Consequently, the predicted excitation energies decrease in βCar-5,8-EPO ($n=9$) and βCar ($n=11$), reproducing well the experimental trends (Fig. 1). In all cases, the LUMO-HOMO energy difference corresponds to the experi-mental values better than the $S_2$ energy, which is typical for this level of theory.

To gain insight into the role of $^1O_2$ in their formation, the oxygenation of βCar was carried out under reduced partial pressure of oxygen or using 1,4-diazabicyclo[2.2.2] octane (DABCO) as the quencher of $^1O_2$, and run in the perdeuterated acetone, followed by HPLC analysis. In all these reactions, highly purified and free from trace impurities samples of βCar were used as the substrate, always freshly obtained by repurification of commercially available pure βCar. The purity of the pigment, also used as a reference in all the analytical runs, is evidenced in Supplementary Fig. 5. In addition, to eliminate possible problems related to the photoexcitation of the solvent in the illumination experiments, red light was applied (>630 nm). To reduce the

partial pressure of oxygen in the reaction medium, extensive purging with high-purity Ar was performed or oxygen was removed after a thorough degassing of the samples under moderate vacuum and then using chemical trapping (Oxoid™ AnaeroGen™ 2.5 L sachets, Thermo Scientific™). The level of residual oxygen was monitored by recording phosphorescence from Pd-pheophytin a (Pd-Pheo) used as the oxygen probe[40]. Clearly, the "solvent isotope trick", useful in $^1O_2$-based organic syntheses[4,43], did (not) work. That is, despite the huge differences in the lifetime of $^1O_2$ in acetone-$D_6$, ~1000 μs vs. ~50 μs in acetone[44], the kinetics of βCar oxygenation and its products in the two solvents were found to be virtually identical (Fig. 2b). In contrast, in the absence of βCar, the kinetics of the self-promoted photodegradation of bacteriopheophytin a (BPheo) is 5-fold faster in the deuterated solvent (Fig. 2a), evidencing at the same time the production of $^1O_2$ in the system. Moreover, regardless of the oxygen level, HPLC analyses of reaction mixtures always reveal the characteristic bands of βCar monoendoperoxides with retention times near 26 min (Fig. 3). Not only is extended Ar purging insufficient to stop the reaction, it even runs under very low oxygen content, or under aerobic conditions when the $^1O_2$ level and its lifetime is strongly suppressed by DABCO. The same results were obtained when Pheo (the $S_1$ maximum at 660 nm), instead of BPheo, was used as PC (Fig. 3). Furthermore, very slow oxygenation of βCar also occurs in the dark, either with or without PC, and white light or even red light above 630 nm accelerates it in the absence of PC, in agreement with a previous study[45]. In all these cases, the same pattern of βCar oxygenation is found (Fig. 3).

The breakdown of βCar-EPOs in organic media was investigated by applying electronic absorption, EPR, time-resolved detection of $^1O_2$ luminescence, LC-MS/MS, and HPLC with in-line spectral analysis. In the dark at ambient temperatures, βCar-EPOs show appreciable stability. For instance, in methanol they keep decomposing slowly, but can easily be detected even after 25 days of standing, if judged using electronic absorption and mass spectroscopies (Supplementary Figs. 1–4), and HPLC (Fig. 1c, d). The changes in the electronic absorption profiles of βCar-7,10-EPO and βCar-5,8,7′,10′-diEPO solutions reflect their slow conversion to species whose absorption maxima are shifted to red, whereas the spectra of βCar-5,8-EPO and βCar-5,8,5′,8′-diEPO practically do not change (Fig. 1c, Supplementary Figs. 1 and 4). The spectral changes of βCar-7,10-EPO in dimethyl sulfoxide (DMSO) are faster and indicate its 1:1 conversion (see the isosbestic points, Fig. 1b) into another pigment with the absorption bands shifted by ~50 nm to the red and showing a new band near 350 nm. Interestingly, there is a strong spin trap effect on the reaction rate, which increases 5-fold in the presence of 5,5-dimethyl-1-pyrroline N-oxide (DMPO) at 150 mM (Fig. 1b).

HPLC analysis shows that in all cases the spontaneous breakdown of βCar-EPOs yields hydrophobic products whose absorption spectra are consistent with the spectral changes (or lack) seen while the βCar-EPOs are kept in organic media in the dark (Fig. 1). Thus, the breakdown product of βCar-7,10-EPO has absorption bands red-shifted with respect to the parental compound, while the spectra of βCar-5,8-EPO and the product of its breakdown are almost identical (Fig. 1c, d). These spectral features of the parental and product molecules agree with the results of the computations (Supplementary Tables 1, 2, and 3). The chromatographic retention times of these products are always much longer (>45 min) than those of the parental EPOs (20-27 min), indicating a change from a relatively high to very low polarity, similar to that of βCar (Fig. 1). In the mass spectra, the formation of these hydrophobic products is correlated with the gradual disappearance of the βCar-EPOs molecular ions (m/z

of ~569 or ~601) and the appearance of new signals of lower m/z values, in particular a signal at 536, which corresponds to that of βCar (Supplementary Figs 1–4). As expected, the stability of βCar-EPOs under illumination is markedly reduced and under white light, they quickly bleach completely.

**Mechanism of the formation of β-carotene endoperoxides**. The participation of $^1O_2$ in the formation of βCar-EPOs is very questionable, as they are formed under a variety of conditions that exclude it from the reaction medium (see below). First of all, at concentrations in the mM range βCar is expected to eliminate (via physical quenching) any $^1O_2$ possibly generated in our system[46]. In fact, the lifetime of $^1O_2$ in it is reduced by three orders of magnitude. Moreover, when using PC, βCar-EPOs do keep being formed under reduced partial pressures of oxygen (using Ar or a chemical absorber of $O_2$) or in the presence of a very high excess of DABCO, an extremely efficient physical quencher of $^1O_2$[30]. Finally, the use of the deuterated solvent has no effect on kinetics and oxygenation products, while PC alone, in the absence of βCar, is degraded much faster than in ordinary acetone (Fig. 2a). Intriguingly, HPLC analyses indicate that wide-range manipulation of the lifetime of $^1O_2$ does not change the βCar oxygenation pattern (Fig. 3), implying that $^1O_2$ does not contribute to it. Such difficulties in halting oxygenation indicate the occurrence of some sort of $^3O_2$ caging in stable contact complexes between open-shell $^3O_2$ and closed-shell βCar molecules[32,47–49].

There are also several arguments that strongly disfavor the concerted mechanism, which specifically requires a planar s-cis-diene conformation[3,4,50]. In the all-trans conformer of βCar this could only be satisfied near the terminal β-ionone rings. In reality, however, these rings are twisted by as much as 46° with respect to the backbone plane (Supplementary Fig. 6), owing to considerable steric interactions between the skeleton and the terminal side methyl groups (CCDC entry No. 1120466)[41,51]. Such a twist causes a significant decoupling of the terminal C = C bonds from the conjugated π-electron system, reflected best in the electronic absorption spectrum of βCar (Fig. 1). Moreover, the computations reveal that the energetic cost of forcing one of the rings to the backbone plane is great and amounts to ~30 KJ/mol (Supplementary Table 1). For this reason, the population of such (half-flat) conformers of βCar at ambient conditions is negligible[52]. Furthermore, the formation of βCar-EPOs with their -O-O- bridge located nearer the molecule center, such as βCar-7,10-EPO, simply cannot be achieved by [2 + 4] cycloaddition. As mentioned above, prior to the reaction, a full conversion of the s-trans to s-cis of the respective single C–C bonds in the βCar skeleton would be required, which is not the case. Considering the alternative mechanisms, one obvious possibility is a thermally activated reaction between the two reactants in their ground states, βCar($S_0$) and $^3O_2$. Indeed, such a reaction takes place in the dark, regardless of the presence/absence of PC, although at a very low rate − the formation of βCar-EPOs becomes noticeable after several weeks of standing at room temperature (Supplementary Fig. 5). Apparently, βCar($S_0$) and $^3O_2$ react very slowly in a spin-forbidden reaction[53]; the vast majority of collisions between the reactants are unproductive, as predicted by reaction rate theory. In the absence of PC, under white light the reaction speeds up somewhat, and surprisingly, even under red light (>630 nm). This light-driven oxygenation seems to be promoted via the $S_2$ state, considering the fact that its activity extends far to the red (Supplementary Fig. 7). However, βCar in singlet excited states, due to both their short lifetimes and Wigner's principle, cannot be expected to directly participate in chemical reactions. Rather, via ISC a longer-lived $^3βCar(T_1)$ will

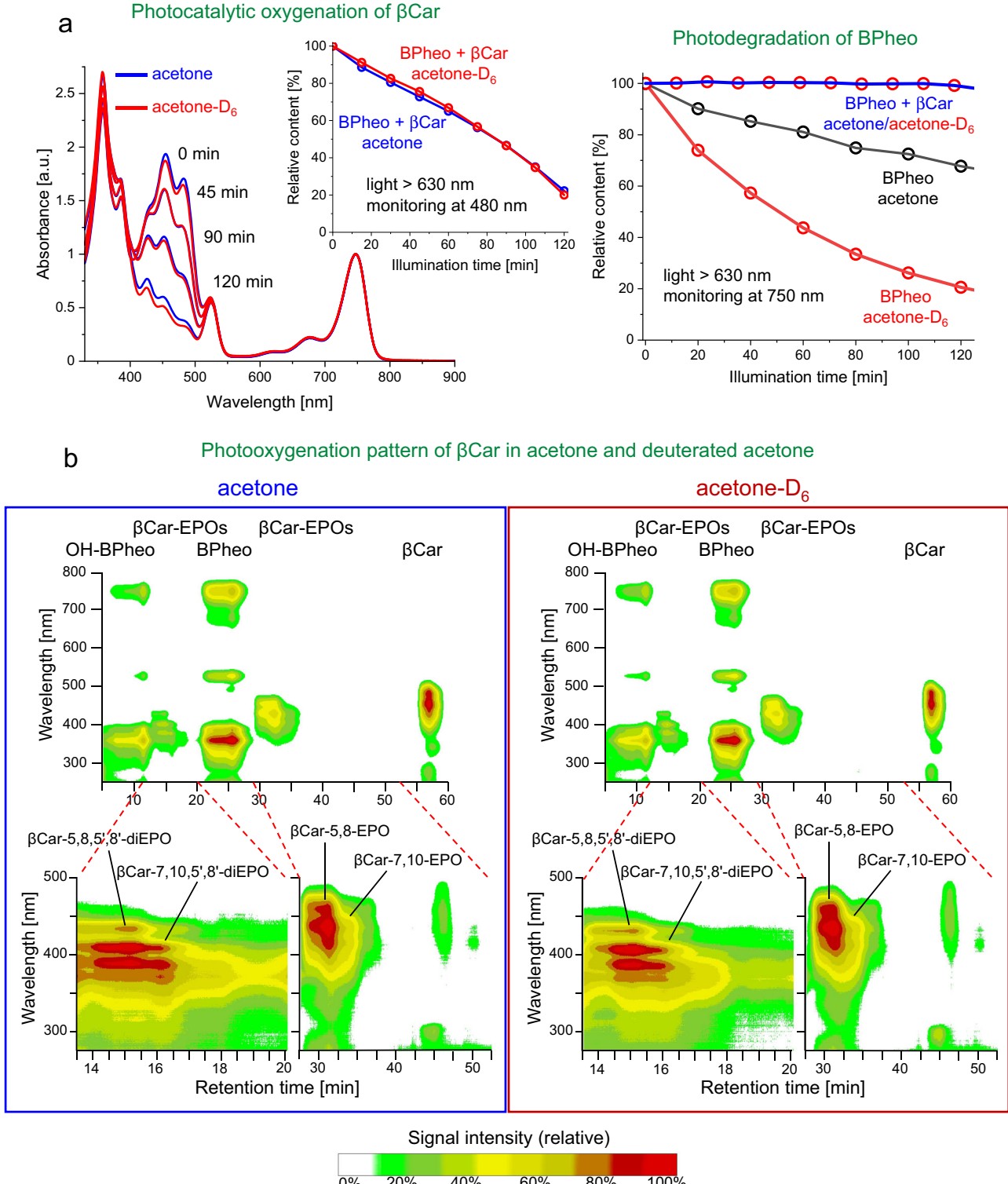

**Fig. 2 Solvent effects on the photocatalytic oxygenation of β-carotene (βCar) and photodegradation of bacteriopheophytin a (BPheo). a** Spectral changes and kinetics (left panel) of the oxygenation of β-carotene (63 μM) in regular and deuterated acetone in the presence of the photocatalyst (90 μM BPheo), and kinetics of auto-photodegradation of BPheo (90 μM) in regular and deuterated acetone in the presence/absence of β-carotene (right panel) under illumination with red light (>630 nm, intensity 370 μmol/m²/s). **b** RP-HPLC analysis of the oxygenation of β-carotene (63 μM) in regular (left) and deuterated acetone (right) in the presence of the photocatalyst (90 μM BPheo). OH-BPheo stands for the 13²-hydroxylated bacteriopheophytin a. Source data are provided as a Source Data file.

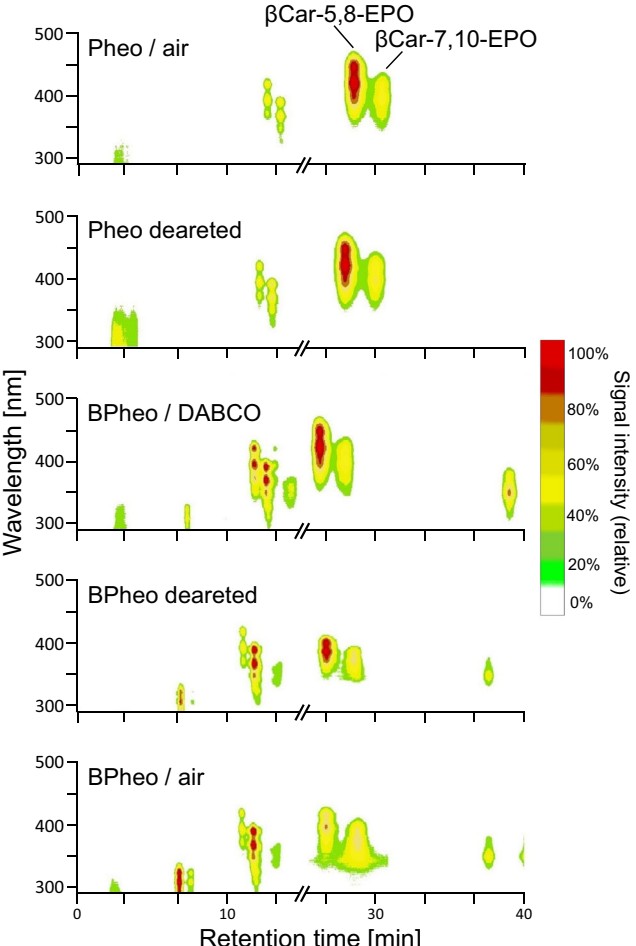

**Fig. 3 Oxygenation pattern of β-carotene under varying partial pressure of oxygen.** Effects of oxygen level and 1,4-diazabicyclo[2.2.2]octane (DABCO) on the photocatalytic oxygenation of β-carotene and formation of its two monoendoperoxides, βCar-5,8-EPO and βCar-7,10-EPO, in acetone, analyzed by RP-HPLC with in-line absorption detection. Pheophytin a (Pheo) or bacteriopheophytin a (BPheo) were used as the photocatalysts and the conditions of the reaction were the same as in the synthesis of the endoperoxides (see Methods for the details).

be populated with a very low quantum yield (~0.001[45,54–56]). Hence, $^3\beta Car(T_1)$ seems to be the factor common to all these light-driven reactions, which is consistent with a strong stimulation by light in the presence of PC. Otherwise, various mechanisms would lead to exactly the same products and the same pattern of βCar oxygenation, which is very improbable. The major ways in which $^3\beta Car(T_1)$ is populated in our system are the following: (i) an efficient EET from $^3$BPheo and $^1O_2$, and (ii) a much less efficient $O_2$-enhanced ISC from the $S_1$ state, resulting in $^1O_2$ sensitization (see below).

The trapping of $^3O_2$ by $^3\beta Car$ occurs in the encounter complex $[^3\beta Car \bullet ^3O_2]^*$ created in collisions between $^3\beta Car(T_1)$ and $^3O_2$ and/or by a direct EET to $[^1\beta Car \bullet ^3O_2]$ from $^3$PC. Although EET requires intimate contact between the donor and acceptor species, the latter path is not improbable, as implied by the occurrence of oxygenation at very low $O_2$ levels. Several factors favor the chemical interaction between reactants in $[^3\beta Car \bullet ^3O_2]^*$: (i) excess energy and long lifetime of $^3\beta Car(T_1)$, within a range of 10–20 μs[57], (ii) the reaction is spin allowed[45]; and (iii) the biradical nature of both $^3\beta Car(T_1)$ and $^3O_2$.

At least two radical-favoring factors are at play here: the allylic-like character of $^3\beta Car(T_1)$ (resonance stabilization) and

stabilization by side methyl groups (Fig. 4a), the effects well known in organic chemistry[49,58]. A substantial electron decorrelation in $^3\beta Car(T_1)$ agrees with the fact that the $^3O_2$ attack sites are located away from the βCar molecule center and mostly 5,8- and 7,10-(mono/di)EPOs are formed. Another chemical reaction of Crts in which the biradical character of their $T_1$ state is clearly manifested is their *cis-trans* isomerization. $T_1$ is a common precursor state for the isomerization of *all-trans* Crts to the *cis* isomers[55,59]. Such a molecular rearrangement necessarily proceeds in several steps: (i) a breakage of a particular C = C bond with a complete loss of electron correlation in a π-bond, invoking a biradical intermediate, (ii) a major rearrangement of this intermediate (a 180° twist around the resulting C–C bond), and (iii) the reformation of the C = C bond[60]. Again, the partial separation of an electron pair towards the ends of the molecule is reflected in the isomer distribution[61].

A covalent bond between $^3O_2$ and $^3\beta Car$ seems to be quickly formed through recombination within $[^3\beta Car \bullet ^3O_2]^*$. This reaction, i.e., oxygen trapping by triplet biradicals[6,14], yields stable biradical peroxy intermediates **1** or **2** (Fig. 4a), and due to the excess energy, it is expected to be barrierless[49] (see below). Intramolecular recombinations in **1** and **2** yield the respective βCar-EPOs. These reactions reveal the surprisingly strong biradical character of $^3\beta Car(T_1)$, which may be partly due to the interactions between paramagnetic species within $[^3\beta Car \bullet ^3O_2]^*$.

Concerning the nature of the encounter complexes formed between βCar and molecular oxygen, even the shorter-lived $^1O_2$ and βCar form $[\beta Car \text{-}^1O_2]^*$ which, after intracomplex EET, converts into $[^3\beta Car(T_1) \bullet ^3O_2]^*$; then it falls apart and $^3\beta Car(T_1)$ relaxes to $^1\beta Car(S_0)$. Most probably, a very rapid formation of the triplet state on βCar in $[\beta Car \text{-}^1O_2]^*$ occurs vertically, that is, on the $S_0$ or near-$S_0$ manifold of this molecule. Such a triplet state with the $S_0$ geometry ("$^3\beta Car(S_0)$") was predicted in large-scale ab initio calculations[62]. This state either relaxes directly on the singlet manifold or relaxes to the native geometry of the triplet state. A single βCar molecule can participate in up to 10,000 of such quenching cycles[5]. $^3\beta Car(T_1)$, generated in this (or any other EET) process, within 10–20 μs relaxes to $S_0$ and is sufficiently long-lived to react chemically, for example, isomerize or form encounter complexes with $^3O_2$.

The computations of the ground state structures indicate that an $^3O_2$ attack on positions 5 or 7 (Fig. 4a) in the large *allylic*-like radical system of $^3\beta Car(T_1)$, which extends from $C_5$ in the ionone ring up to $C_{10}$, is more probable than an attack on positions 8 or 10, due to the higher stability of **1** and **2** (Supplementary Table 1). The structures of the final products imply the sequence of steps leading to the closure of the -O-O- bridges. As no rearrangement around $C_5$ is possible, βCar-5,8-EPO must be formed while retaining the original conformation. In contrast, the formation of the 7,10-$O_2$ bridge requires an 8-s-*trans* to 8-s-*cis* change in geometry at the $C_8$–$C_9$ bond. This is achieved by the rehybridization ($sp^2 \rightarrow sp^3$) of two respective carbon atoms and two rotations around the C–C bonds (see Fig. 4a). Due to these extra steps, the overall rate of the 7,10-product formation is necessarily slow, which is highly consistent with the reaction yield, which is lower by a factor of 3–4 with respect to the 5,8-product. The 7,10-product is energetically less stable than the 5,8-product, by ~44 kJ/mol, and a major contribution to this value comes from the steric interactions between the $O_2$ bridge and the skeleton (Supplementary Table 1). This strain is rather confined because of the protective effect of side methyls on the skeletal conformation[41]. The computations on the truncated βCar-EPOs, which show deviations from the $sp^2$ geometry on $C_{11}$ after the methyl group has been removed, confirm this notion (Supplementary Table 2). Concerning the driving force for oxygenation, a

**Fig. 4 The formation and breakdown of β-carotene endoperoxides. a** The mechanistic details of the formation of β-carotene endoperoxides as a reaction between two triplet biradicals: $^3O_2$ and $^3βCar(T_1)$. The 5,8-product (βCar-5,8-EPO) results from the attack of $^3O_2$ on $C_5$, followed by a rearrangement to a conjugation-stabilized allylic radical **1** and then a closure of the endoperoxide ring via intramolecular radical recombination. The 7,10-product (βCar-7,10-EPO) results from the attack on $C_7$, during which the hybridization of this carbon atom changes from $sp^2$ to $sp^3$, followed by two 180° rotations around C–C bonds, as indicated, leading to the stabilized allylic-like radicals **2** and **3**, respectively; the recombination of **3** via intermediate **4** yields βCar-7,10-EPO. **b** The pathway of light-promoted βCar-EPOs' decomposition that specifically leads to the release of $^1O_2$ and formation of β-carotene isomers that have, respectively, 9 and 11 double C–C bonds conjugated in the main π-electron system. This involves cleavages of the C–O bonds, yielding peroxide- and allylic-type biradical species. Intermediate **4** appears to be common to both the formation and the breakdown of βCar-7,10-EPO. The stereochemistry and the hydrogens are explicit to indicate the stereochemical course of the reactions. 5,5-dimethyl-1-pyrroline N-oxide (DMPO) was used only in the dark reaction to chemically trap the free radical intermediates. The green arrow indicates the suggested site of a DMPO attack. **c** The pathways of light-induced fragmentation of βCar-7,10-EPO into β-cyclocitral and apo-10'-carotenal, the products of β-carotene oxygenation identified in photosynthetic tissue in plants. The fragmentation involves the photolysis of the O–O bridge into a biradical species and their consecutive rearrangement and cleavage into a cascade of lower mass free (bi)radicals. Alternatively, initially a Kornblum–De La Mare rearrangement takes place, leading then to the same products.

simple comparison of the changes in the reactants/products total energies in the $O_2$−βCar−βCar-EPO system show that it is greater for the 5,8-product (Supplementary Table 1). Thus, the formation of βCar-EPOs is under both kinetic and thermodynamic control, favoring βCar-5,8-EPO as the main product, which is consistent with the experiment. Additionally, in vivo βCar-5,8-EPO is the major product of βCar oxygenation and its accumulation in light-stressed photosynthetic tissues is well documented[23,24]. In contrast to our conclusions (see below), it is regarded as an early index of $^1O_2$ production in leaves, and its low-mass breakdown products, in particular β-cyclocitral and apo-10'-carotenal, are considered to be markers of oxidative stress, which participate in stress signaling and the induction of acclimation genes[23–25]. In Fig. 4, the plausible pathways of light-induced βCar-EPO decomposition into $O_2$ and carotenes (Fig. 4b)

and the breakdown to low-mass products (Fig. 4c) are shown. βCar-7,10-EPO, owing to its photolability and the retention of oxygen atoms on $C_7$ in the former and on $C_{10}$ in the latter, appears to be their parental molecule (Fig. 4c). The βCar-7,10-EPO breakdown begins with a Kornblum–De La Mare rearrangement favored by protic media[58], followed then by photochemical processes or, alternatively, with a photolytic cleavage of the -O-O-bridge. In either case, an *allylic*-like 1,4-biradical intermediate is formed, additionally resonance-stabilized by a large π-electron system. This intermediate in the excited state cleaves into a variety of lower mass free radical and biradical products in a Norrish type II reaction[49,58,63]. This mechanism explains the formation of β-cyclocitral and apo-10'-carotenal as the end products. Consequently, apo-8'-carotenal is expected to be among the products of βCar-5,8-EPO breakdown.

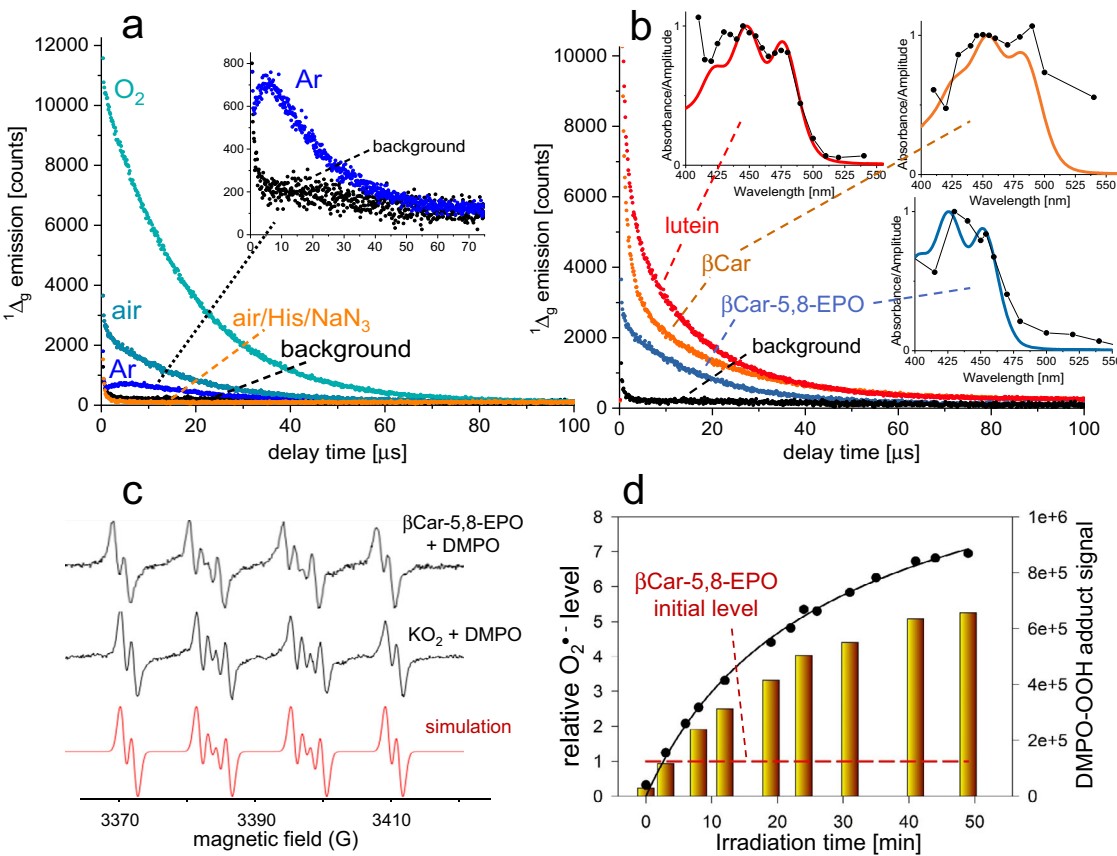

**Fig. 5 Generation of reactive oxygen species from β-carotene endoperoxides. a** Time-resolved detection of luminescence of singlet oxygen that is generated by βCar-5,8-EPO (4 μM) excited with laser flashes at 454 nm in acetone, under various oxygen tensions and in the presence of singlet oxygen quenchers, histidine (His) and NaN₃. Inset: expansion of the signal detected under Ar. **b** Singlet oxygen sensitization by lutein, β-carotene and βCar-5,8-EPO, each at 4 μM in acetone solutions. In the insets, the corresponding action spectra of singlet oxygen sensitization by the pigments are shown. **c** the 5,5-dimethyl-1-pyrroline *N*-oxide (DMPO)-spin trapping of free radicals generated by βCar-5,8-EPO under blue light or $KO_2$ in dimethyl sulfoxide; upper panel—the EPR spectrum of DMPO-OOH adduct generated after 30 min of irradiation (2 mW/cm²) of βCar-5,8-EPO (35 μM); middle panel—the EPR spectrum of DMPO-OOH adduct generated in the $KO_2$ solution in the dark; bottom panel—the spectrum simulated by WinSim program using the following parameters: $a_N = 12.83$; $a_H^\beta = 10.36$, $a_H^\gamma = 1.32$ G. No DMPO-OOH adducts were detected in neat dimethyl sulfoxide, nor in the solution of βCar-5,8-EPO in dimethyl sulfoxide in the dark. **d** The accumulation of the DMPO-OOH signal and the corresponding rise in concentration of $O_2^{•-}$ detected during irradiation of βCar-5,8-EPO (35 μM) in dimethyl sulfoxide with blue light (402–508 nm, 11.6 mW/cm²). Source data are provided as a Source Data file.

**Generation of reactive oxygen species by β-carotene endoperoxides**. The abilities of βCar-EPOs to quench and release ¹O₂ were evaluated using the nanosecond time-resolved detection of ¹O₂ luminescence in the NIR (1270 nm). Measurements were taken in acetone and βCar was used as a reference quencher. In the quenching experiment, ¹O₂ was produced using rose Bengal (RB, 2 μM) as PC (excitation at 590 nm). The stability of the pigments under the measurement conditions was assessed by their irradiation with 454 nm laser pulses at 1 kHz. In the presence of βCar or βCar-5,8-EPO (each 2.66 μM), the lifetime of ¹O₂ luminescence ($τ_Δ$) drops from ~50 μs in neat acetone to 15 μs and 23 μs, respectively. In the latter case, $τ_Δ$ extends upon prolonged irradiation, reaching about 47 μs along the pigment bleaching (Supplementary Fig. 8). Under the same conditions, $τ_Δ$ in βCar solutions is almost constant. Furthermore, irradiating βCar-5,8-EPO solution with 590 nm pulses does not affect $τ_Δ$.

¹O₂ luminescence was recorded in acetone solutions of βCar-5,8-EPO, saturated with air, O₂, or Ar, and excited at 454 nm. Under Ar, a weak but clear ¹O₂ signal is seen (Fig. 5a), which can be explained by the release of ¹O₂ from βCar-5,8-EPO upon illumination. This seems to be an uncommon example of the release of ¹O₂ from a nonaromatic EPO[3,4]. Intriguingly, however,

the signal rise is non-instantaneous and its maximum amplitude occurs only after several μs. The action profile of ¹O₂ generation matches the spectrum of βCar-5,8-EPO (Fig. 5b), while histidine and NaN₃, extremely efficient chemical/physical quenchers of ¹O₂, greatly reduce the signal amplitude, thus providing evidence for the origin of ¹O₂. Under aerobic conditions and particularly under O₂, the signal of ¹O₂ luminescence strongly increases and its rise is instantaneous (Fig. 5a). These differences in the amplitudes and decay profiles indicate different mechanisms of ¹O₂ generation in the absence and presence of O₂. Evidently, photoexcited βCar-5,8-EPO releases ¹O₂ stepwise in a slow reaction and ¹O₂ must diffuse off to emit luminescence (the product is a ¹O₂ quencher). In contrast, the shape of the signal and the strong effect of O₂ indicate that βCar-5,8-EPO is also able to sensitize ¹O₂, which is surprising. A comparative approach, using Chla as a reference[39], showed the quantum yield of this process ($ϕ_Δ$) to be around 0.5%. Obviously, $ϕ_Δ$ is the net value which takes into account the quenching of ¹O₂. This finding led us to examine whether such a ¹O₂ sensitization property is shared by other Crts, βCar, and lutein. Since we were aware of the risk of ¹O₂ sensitization by trace contaminations, precautions were taken to prepare the pigments in a strictly Chl-free regime (synthetic

βCar, virgin glassware, quartz cuvettes and HPLC columns, lutein was of natural origin, showing virtually null emission of Chl fluorescence). To our surprise, both Crts reveal similar $^1O_2$ sensitizing properties, as evidenced by the action spectra (Fig. 5b), which, importantly, peak around 450 nm and differ very much from the action spectra obtained for Chls in our experimental setup.

To gain more insight into the mechanism of photodecomposition of βCar-EPOs, the EPR measurements were taken using DMPO as a spin trap. The irradiation of βCar-5,8-EPO (35 μM in DMSO) with blue light (400–500 nm) produces a four-line EPR spectrum with hyperfine splittings characteristic of the DMPO-OOH spin adduct (Fig. 5c), that is, the product of $O_2^{\bullet-}$ trapping[64].

During the first minutes of irradiation, the signal accumulation rate is the highest, but $O_2^{\bullet-}$ can still be detected even after 40 min of irradiation (Fig. 5d). The appearance of $O_2^{\bullet-}$ is consistent with the free radical mechanism of $O_2$ release from βCar-EPOs (see below) and with the dark reaction rate-enhancing effects of both DMSO, known to stabilize free radicals, and DMPO in particular, which seems to literally abstract $O_2^{\bullet-}$ from the substrate and pushes the reaction forward. In an attempt to estimate the levels of $O_2^{\bullet-}$ photo-released from βCar-5,8-EPO, $KO_2$ in DMSO was used as the source of $O_2^{\bullet-}$[65,66]. The spectra of the species generated from $KO_2$ and by irradiated βCar-5,8-EPO are almost identical (Fig. 5c). The signal that builds up over 30 min of irradiation corresponds to that of 100–150 μM $O_2^{\bullet-}$ as attained in 1.5 mM $KO_2$. Apparently, this value far exceeds the endoperoxide concentration used in the experiment (35 μM). Such an efficient production of $O_2^{\bullet-}$ can be accounted for by the photochemical reactions analogous to those discussed above for βCar-7,10-EPO and outlined in Fig. 4c, leading to a cascade of lower molecular mass free radical products that can react with $O_2$ to give $O_2^{\bullet-}$[49].

To verify the biological relevance of the release of free radicals from βCar-EPOs, the production of free radical species was evaluated in liposomes under illumination using the EPR technique. Initially, the DMPO-OH adduct was detected (Supplementary Fig. 9) as the product of a spontaneous decay of DMPO-OOH in an aqueous milieu. Upon prolonged illumination, a clear EPR spectrum of the DMPO-CH₃ adduct was recorded, as a result of the reaction of DMPO with carbon-centered radicals, most likely secondary ones, generated in the lipid environment due to the cascade of (primary) free radicals triggered along βCar-EPO decay. This result supports the free radical mechanism of βCar-EPO photodegradation and the pro-oxidant properties of βCar-EPOs.

**Mechanism of oxygen release from β-carotene endoperoxides**. The release of $O_2$ from the βCar-EPOs, both in the dark and under illumination, yields isomeric carotenes of molecular masses equal to that of βCar (m/z 536). The product of $O_2$ release from βCar-7,10-EPO retains the parental conformation of EPO and the system of 11 conjugated C=C bonds (Fig. 1), but each different from that in βCar. On the basis of its experimental and calculated electronic absorption (Supplementary Table 1), and in particular on the appearance and intensity of the "cis" band in the UV range (Fig. 1b), in consistency with our vectorial model of the $S_2$–$S_n$ interaction[51], it can be identified as 8-s-cis-β,β-carotene. In the case of βCar-5,8-EPO, the product π-electron system is split into two separate sets of 9 and 2 (as a diene) conjugated C=C bonds, which is reflected as the blue shift in its absorption maxima (Fig. 1d). Its characteristics and the similarity of the absorption spectrum to that of 7,8-dihydro-βCar[67] allow one to identify it as 6-dehydro-8-hydro-γ,β-carotene (Fig. 4). The spectral features of both products are consistent with the computational results; their

total energy is higher by 30-50 kJ/mol with respect to βCar (Supplementary Table 1), which explains in part their low stability.

The two products constitute peculiar examples of carotenes: one having a non-canonical s-cis configuration and the other a split π-electron system[67]. The carotenes of such unusual structures, especially the latter, cannot be the products of any concerted reactions that are very often considered in the breakdown of various EPOs[4,9]. Rather, as shown in Fig. 4b, their structures imply that $O_2$ is released stepwise in radical reactions which somewhat resemble the reverse oxygenation pathway (Fig. 4a). In the initial step of the light-induced process, cleavage of the C–O bond yields the (excited) biradical species, analogous to those that appear during EPO formation. In the next step, an $O_2$ molecule, most probably $^1O_2$, is released from the peroxy moiety[68], with a concomitant recovery of 11 C=C bonds. Clearly, the formation of 6-dehydro-8-hydro-γ,β-carotene, via a hydroperoxy (bi)radical intermediate, involves an intramolecular abstraction of H atom. Analogous (bi)radical hydroperoxyl appears in the photocleavage of ketones according to the Norrish type II mechanism[58]. The release of $O_2$ is then synchronized with the migration of the H atom to the $C_8$ site. As discussed above, these slower steps necessarily reduce the overall rate of the reaction, which is manifested as the non-instantaneous rise in the $^1O_2$ luminescence signal (Fig. 5a), different from the characteristic rise and exponential decay of the $^1O_2$ signal following sensitization (Fig. 5b). Such a radical mechanism explains the formation of two separate π-electron systems in the case of βCar-5,8-EPO and a single one with cis conformation in the case of βCar-7,10-EPO. In neither case is the parental structure of all-trans β-carotene retrieved.

The occurrence of biradical intermediates during $O_2$ release, such as **4** in Fig. 4b, is consistent with the unexpectedly high amounts of free radicals generated during the photoinduced breakdown of βCar-EPOs. According to our estimation, the amounts of DMPO-trapped free radicals are higher than the amount of βCar-EPO used in the experiment by almost an order of magnitude (Fig. 5d). Apparently, photolysis of βCar-EPOs triggers an avalanche of free radical byproducts, which would then account for the pro-oxidant activity of βCar as mediated by its EPOs, even in the absence of external PC. In addition, βCar-EPOs may themselves sensitize $^1O_2$ (Fig. 5b), which can be an additional source of some free radicals.

Intriguingly, because the same carotene products are formed in the dark and upon illumination, the mechanism of $O_2$ release appears to be independent of the way it is activated, either thermally or photochemically. This seems to be an uncommon case in which the photochemical and thermal pathways merge at some point and lead to the same product(s)[49,69]. The photochemical reaction is likely to be non-adiabatic (Fig. 6), which speaks in favor of the (bi)radical mechanism of $O_2$ release[69]. Due to a promptly occurring major reorganization of the photochemically populated transition state TS*, the energy level of the primary photoproduct (PPP), which constitutes a relaxed βCar-EPO biradical (**1** or **4**), falls below the level of the thermally activated transition state $TS_\Delta$, which renders the reaction irreversible. On the other hand, the energy level of $TS_\Delta$ must be quite high, which explains the considerable thermal stability of βCar-EPOs.

**Photocatalytic generation of $^1O_2$ by carotenoids**. As mentioned above, Crts are excellent acceptors (quenchers) of excitation energy from $^1O_2$. Our discovery of $^1O_2$ sensitization by Crts shows that EET in the reverse direction, from Crt* to $^3O_2$, is also possible, although with a much lower efficiency (~0.005). To our

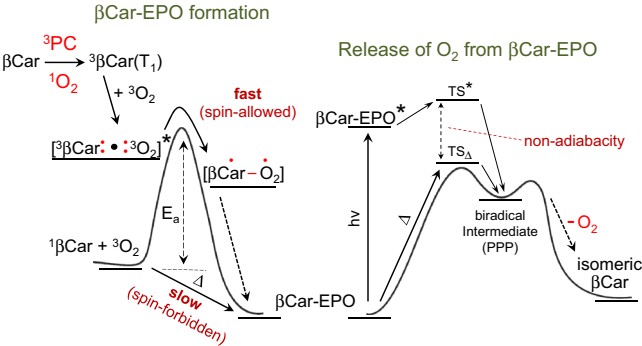

**Fig. 6 Intermediates in the formation and breakdown of β-carotene endoperoxides.** The key steps in the non-concerted formation and release of $O_2$ from β-carotene endoperoxides (βCar-EPO). The photocatalyst (PC)-promoted EPO formation is a fast spin-allowed, probably barrierless (the activation energy $E_a$ close to zero), reaction between the ground state oxygen and β-carotene (βCar) in the triplet state, which involves biradical intermediates (Fig. 4a). The release of an oxygen molecule during the breakdown of EPO also proceeds through an intermediate of free radical character (Fig. 4b). The release of $O_2$ can be promoted either thermally (slow) or photochemically (fast) and the two pathways that via respective transitions states (TS* and $TS_\Delta$) lead to the same isomeric β-carotene, converge at some point. Due to the excess excitation energy, the photochemical pathway is non-adiabatic.

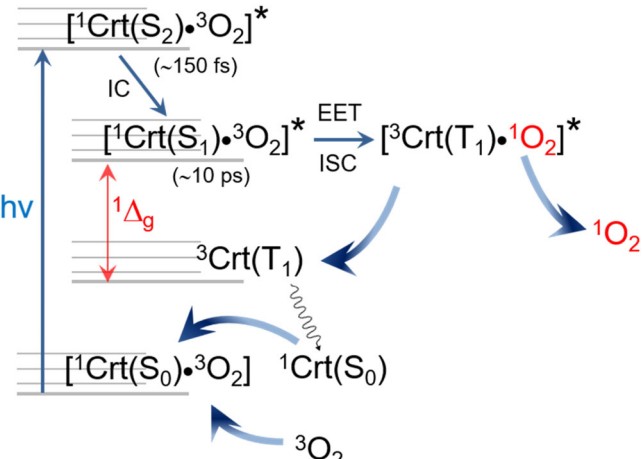

**Fig. 7 Sensibilization of singlet oxygen by carotenoids in solution.** The photocatalytic generation of $^1O_2$ by carotenoid excited from its ground state ($^1Crt(S_0)$) to the $S_2$ state ($^1Crt(S_2)$) takes place in the encounter complex with the ground state oxygen, $[^1Crt(S_2)\bullet^3O_2]^*$. Excitation energy transfer (EET) to $^3O_2$ occurs in this exciplex with the intersystem crossing (ISC) from the $S_1$ state of the pigment, populated via internal conversion (IC) from $S_2$, and results in an ultrafast double spin exchange. The red arrow indicates the energy difference between the carotenoid $S_1$ state ($^1Crt(S_1)$) and $T_1$ state ($^3Crt(T_1)$) that matches exactly the energy of singlet oxygen ($^1\Delta_g$), which is the prerequisite for the effective quenching mechanism.

knowledge, direct sensitization of $^1O_2$ by Crts has never been reported before, and it sheds some new light on the puzzling photophysics of these pigments. On the other hand, there have been some indications for such Crts activity obtained using biochemical methods[70,71], whereas the simultaneous generation and quenching of $^1O_2$ by a single species is a known phenomenon. For instance, melanins in solution both generate and quench $^1O_2$[72]. At the moment it is difficult to be sure about the mechanism responsible for the EET from Crt* to $^3O_2$ in the excited contact complex $[Crt^*\bullet^3O_2]$ and it deserves further investigation. The experiments with oxygen removal indicate that such relatively long-lived complexes between the two species in their ground states pre-exist in solution and, most probably, the excitation of Crt occurs within its complex with $^3O_2$ (Fig. 7). Three symmetry-allowed electronic states of $^1βCar(S_2)$, must be considered to be the excitation energy donors to $^3O_2$: the initial "bright" $S_2$, a "dark" $S_1$, and a low-energy $T_1$[73,74]. The latter state may, in principle, be produced directly via ISC from $S_2$, and such a relaxation pathway has been observed, e.g., in bacterial LH complexes[34], while in $[^1βCar(S_2)\bullet^3O_2]^*$ ISC to $T_1$ may be $O_2$-enhanced[73]. However, in Crts in solution the efficiency of ISC is extremely low, ~0.001[54,55] and $T_1$ itself is a product of $^1O_2$ quenching, and therefore this state can also be ruled out on both the energetic grounds and spin statistics unfavorable for EET to $^3O_2$[48]. The sensitization of $^1O_2$ from the singlet excited states is well known[49,73,75]. Hence, Crt could directly transfer the excitation energy to the energetically nearest state of $^3O_2$, namely $^1\Sigma_g^+$ (~13100 $cm^{-1}$), followed by $^1\Sigma_g^+ \rightarrow ^1\Delta_g$ IC[44]. Nevertheless, owing to the extremely fast $S_2 \rightarrow S_1$ IC (~150 fs[56]), a direct population of $^1\Sigma_g^+$ from $S_2$ also seems very unlikely. Instead, EET to $^3O_2$ may rather occur from a longer-lived $S_1$ (~10 ps[56]), energetically located above $^1\Delta_g$[74], and $^1O_2$ is then produced via an ultrafast double spin exchange in $[^1βCar(S_1)\bullet^3O_2]^*$. In effect, Crt relaxes from $S_1$ to $T_1$ and $[^3βCar(T_1)\bullet^1O_2]^*$ falls apart. After $T_1$ relaxation, the photocatalytic cycle may act again, as depicted in Fig. 7. Two critical factors, the $S_1$-$T_1$ splitting in Crt, in particular βCar, which matches the $^1\Delta_g$ energy perfectly, and the enhancement of

spin exchange due to the presence of paramagnetic species in the collision complex $[^1βCar(S_0)\bullet^3O_2]$, favor the above mechanism. Furthermore, EET from $S_1$ populated via $S_2$ excitation functions well in the LH antenna[34,76]. Clearly, both the very low populations and the short lifetimes of $S_1$ as well as $[^1βCar(S_0)\bullet^3O_2]$ limit the quantum yield of the entire process.

**Implications for biological systems.** The weak photosensibilization of $^1O_2$ by Crts, with their $\phi_\Delta$ value near 0.005, does not seem to have a major impact on how these pigments function in biological systems. Nevertheless, it provides new information about their complex photophysics and merits more attention. On the other hand, our findings shake up the prevailing view of the role of $^1O_2$ as a chemical trigger in oxidative stress signaling and the role of βCar as an anti-oxidant/pro-oxidant and (photo)protectant. In addition, the photodegradation of βCar-EPOs leads to the release of $^1O_2$ and a cascade of free radicals that may impair lipids, which explains the pro-oxidant activity of βCar and its derivatives.

The fact that βCar-EPOs are not the products of a reaction with $^1O_2$ is of great relevance to natural photosynthetic systems. Most importantly, the endoperoxides of βCar are not markers of oxidative stress due to $^1O_2$, but of the overproduction of extremely hazardous Chl triplets in photosystems quenched by βCar. In principle, the pool of $^3βCar$ may also partly originate from the physical quenching of $^1O_2$, but usually the levels of the latter are very low[77,78]. The use of a deuterated solvent shows that in our model system the contribution of this path is indeed negligible. Hence, overexcitation signaling from the photosynthetic apparatus appears to be based entirely on the $^3Chl$-$^3βCar$ relay and the reactivity of long-lived $^3βCar$, rather than on short-lived $^1O_2$, in contrast to how it is currently viewed. A model of the protective functioning of βCar and the role of βCar-EPOs in stress signaling in the photosystems, that takes into account these findings, is schematically depicted in Fig. 8. βCar-EPOs, the products of the reaction between $^3βCar$ and $^3O_2$, in lipid membranes are able to diffuse to sites that are far from their own origin, and their breakdown products may act as markers in stress

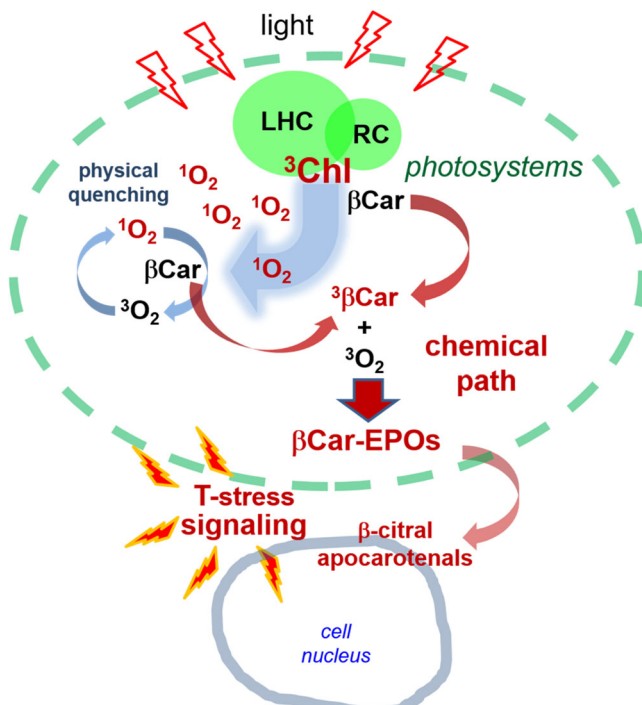

**Fig. 8 Triplet state-stress signaling in photosynthetic apparatus.** The model of photoprotective functioning of β-carotene (βCar) and the role of βCar-endoperoxides (βCar-EPOs) and their breakdown products in signaling of triplet overproduction within the photosystems is shown. The lifetime of $^1O_2$ in the photosynthetic membranes is greatly reduced due to the physical quenching by βCar and other quenchers. Under ambient or moderately strong light, the triplets of chlorophylls ($^3$Chl) are efficiently extinguished by xanthophylls and other carotenoids, including βCar. Under strong illumination, $^3$βCar, generated in excess, chemically reacts with ground state molecular oxygen ($^3O_2$) and undergoes oxygenation to βCar-EPOs. The low molecular mass products of their photochemical decay diffuse to the nucleus and trigger cellular protective and repair mechanisms, acting as the triplet state-stress (T-stress) signals.

signaling and activation of stress-related genes. Although βCar-5,8-EPO is the chief product of βCar oxygenation, also in vivo[16,18,22], a less stable βCar-7,10-EPO appears to be a more natural source of β-cyclocitral and apo-10′-carotenal, two of such markers.

The $^1O_2$-independent triplet state relay makes more sense in biological systems that evolved to effectively cope with ROS. It not only circumvents the barrier of $^1O_2$ quenchers, but also conveys more specific information about triplet state overproduction. Sophisticated triplet sensing, via the T–T reaction and its products, may activate, in a pre-emptive way, various cellular protective and repair mechanisms. In a sense, βCar oxygenation in the photosynthetic apparatus constitutes a smart safety valve that controls the $^3$Chl population while conveying chemical information. Furthermore, considering the high rates of βCar-EPOs' photodecomposition, their accumulation in photosynthetic tissues implies that the efficiency of the $^3$βCar(T$_1$) + $^3O_2$ → βCar-EPOs reaction in vivo is higher than anticipated[5,22].

In light of our findings, it is important to understand how the photoprotective mechanisms present in the photosynthetic apparatus deal with triplet overproduction. Most of βCar in the photosystems is hosted by the PSII core, and this is also the primary site of its oxygenation to βCar-EPOs. This and other degradation processes occur permanently, as a continuous turnover of βCar, as well as of Chls but not xanthophylls, takes

place in the photosystems, even under relatively low light[79]. Whereas, the major antenna, LHCII, exclusively binds xanthophylls (lutein, neoxanthin, and/or zeaxanthin/violaxanthin) which appear not that much prone to oxygenation[22], being already partly oxygenated. The overexcited Chls in LHCII are a potent source of triplets and $^1O_2$, and several photoprotective mechanisms evolved to eliminate or minimize damage. In the first place, the triplets are efficiently quenched by the L1/L2 luteins, which are located in close proximity to the Chls. Whereas, neoxanthin acts as an oxygen barrier, limiting the access of $O_2$ to the inner domain of the complex, thereby strongly contributing to the photostability[80]. Direct evidence for the limited access of $O_2$ to lutein sites was inferred from the decay kinetics of their T$_1$ state[81]. A comparison of the molecular packing within the structures of LHCII and PSII confirms this notion, showing differences in the exposure of βCar and the xanthophylls (Supplementary Fig. 10). The LHCII assembly is a very tightly packed structure in which the luteins are buried deeply inside the protein domain. Neoxanthin locks the entire assembly and partly protrudes into the lipid bilayer[82]. Intriguingly, the protruding fragment consists of the 6-8 allenic and 3,5-dihydroxy moieties that protects it from any $O_2$ attack. Altogether, on the one hand, very little triplets are likely to "escape" from overexcited LHCII, and on the other hand, $O_2$ is not allowed to penetrate into the sites where the triplets are produced. Under strong light, when the capacity of these photoprotective mechanisms is exceeded, the population of unquenched triplets rises and deleterious $^1O_2$ is produced. Then the other xanthophylls bound in LHCII and βCars in the core antenna and around the P700 and P680 reaction centers, as well as other compounds abundant in the photosynthetic membranes, e.g., tocopherols and prenylquinols, take over and physically scavenge $^1O_2$[83–86]. In effect, the concentration of $^1O_2$ in the photosystems is kept at a very low level[87] and its residence lifetime is shortened from tens of microseconds (in solution, Fig. 5) to tens of nanoseconds[77,78]. Thus, in practical terms, any 'chemistry' involved in $^1O_2$ elimination in this environment is quite limited, but appears to take place and pigments are continuously degraded while their degradation products act as "T-stress" signals (Fig. 8).

In conclusion, our study reveals that the chemical reactivity of β-carotene, the model carotenoid that is responsible in part for its functioning as anti-oxidants and photoprotectants, is triplet state-driven. The first excited triplet state of β-carotene, $^3$βCar(T$_1$), plays a pivotal role in its oxygenation to endoperoxides, showing a remarkable biradical character in reactions with another species of biradical nature, $^3O_2$. In the absence of $^1O_2$, light strongly stimulates these reactions, particularly in the presence of PCs, which points to the crucial role of $^3$βCar(T$_1$). We find no indications of the chemical reaction between $^1$βCar(S$_0$) and $^1O_2$, and hence βCar-endoperoxides are not the products of the chemical quenching of $^1O_2$ by βCar, as is commonly thought. On the contrary, it is βCar in the triplet state that is chemically quenched by $^3O_2$ in a very specific type of T–T annihilation reaction.

βCar-endoperoxides are formed via non-concerted recombination of two biradicals: $^3O_2$ and an *allylic*-type excited biradical $^3$βCar(T$_1$), resulting in a nearly barrierless closure of the -O-O-bridges. The breakdown of βCar-endoperoxides follows in the reverse direction an analogous non-concerted radical mechanism. Such radical mechanisms enable intramolecular rearrangements that account for the structures of the products. Under ambient conditions in the dark, βCar-endoperoxides are surprisingly stable, release $O_2$ and quantitatively revert slowly to carotenes—isomers of βCar. βCar-endoperoxides are much more photolabile and readily release $^1O_2$ and free radicals. To our knowledge, they are a rare example of non-aromatic endoperoxides that release $^1O_2$. The

yields of the liberated free radicals are very high, most likely due to a photochemically generated cascade in the Norrish type II reactions, and may lead to the damage of lipids. The production of various ROS by βCar-endoperoxides explains well the controversial pro-oxidant activity of βCar and the paradoxical switch of its anti- to pro-oxidant features under higher tensions of $O_2$. At low $O_2$ levels, βCar keeps up as the protector, whereas at higher levels, the oxygenation to βCar-endoperoxides, and their breakdown into $^1O_2$ and free radicals, become dominant. We have also found that Crts not only physically quench $^1O_2$, but are also able to photo-catalytically generate $^1O_2$, though with low efficiency. We suggest that the essential step in this photocatalysis comprises excitation energy transfer to $^3O_2$ from Crt* in the $S_1$ state that produces $^1O_2$, and occurs with a concomitant transition from the $S_1$ state to the $T_1$. The sensitization is facilitated by the exact matching of $^1\Delta_g$ energy to S-T splitting in Crts. These findings shed new light on the sophisticated photophysics and photochemistry of these fascinating isoprenoid chromophores.

## Methods

**Synthesis and isolation of β-carotene endoperoxides.** The endoperoxides of β-carotene were synthesized photocatalytically from synthetic all-*trans* β-carotene (Sigma, Germany) and isolated following the methods described earlier[17,18]. Briefly, BPheo, prepared from pure bacteriochlorophyll a via demetalation, was used as a PC activable by red light in the range above 630 nm. A mixture of β-carotene (typically 40–60 μM) and BPheo/Pheo (90 μM) in acetone (60 ml) was illuminated with red light (>630 nm) at the intensity of 2300 μmol/m²/s while stirring for 90 min at room temperature. Afterwards, the solvent was removed under reduced pressure and the residue was subjected to high performance liquid chromatography on a LiChroCART 250-4 LiChrospher HPLC Cartridge 100 RP-18 (5 μm) using a ProStar 230 system (Varian, USA) coupled with a diode array detector (J&M Tidas, Germany). The separations were achieved using a gradient composition of acetonitrile, methanol and tetrahydrofuran. If necessary, the fractions of endoperoxides were subjected to 2nd and 3rd rounds of purification under the same conditions. The dried products were stored under Ar at −80 °C in the dark. The absorption spectra of the pigments were measured on Cary 400 and Cary 60 spectrophotometers (Varian, USA).

β-Carotene used as the oxygenation substrate in the analytical runs was obtained by repurification of commercially available pigment of HPLC purity (Sigma, Germany). The repurification was done by a reversed-phase HPLC on a LiChroCART 250-4 LiChrospher HPLC Cartridge 100 RP-18 (5 μm) using the same solvent system as the one used in the isolation of β-carotene endoperoxides. The purified pigment was stored until use under Ar at −80 °C in the dark.

**Oxygenation of β-carotene under reduced oxygen partial pressure.** The oxygen was removed from the samples by a thorough degassing under reduced pressure (5 mbar) followed by either extensive purging with high-purity Ar (99.999%) or extensive (3–4 days) chemical trapping with the use of OxoidTM AnaeroGenTM 2.5 L sachets purchased from Thermo Scientific (USA). The oxygen level was monitored by detecting the phosphorescence of Pd-Pheo[40]. The oxygenation reactions were run in acetone as described above using β-carotene at 40 μM, BPheo at 70 μM and under red light (>630 nm) at the intensity of 1100 μmol/m²/s. The same conditions were used in the reactions run in perdeuterated acetone (Sigma, USA). DABCO was used at the concentration of 9 mM.

**Mass spectroscopy.** The full MS spectra were collected by direct injection of aliquots into the MS spectrometer (LCQ Fleet Ion Trap Mass Spectrometer), Dionex UltiMate 300 (Thermo Fisher Scientific, USA). The stability of the pigments has been tested in methanol, acetonitrile and acetone. In the latter case, due to poor ionization, before the injection the solvent was evaporated and the sample redissolved in acetonitrile.

*UHPLC-DAD-HESI-MS/MS analysis.* The mass analysis was carried out using a liquid chromatography system (UHPLC) consisting of a quaternary pump with a degasser, a thermostated column compartment, an autosampler and a diode array detector connected to a LCQ Fleet Ion Trap Mass spectrometer, equipped with a heated electrospray ionization (HESI). The Xcalibur (version 2.2 SP1.48) and LCQ Fleet (version 2.7.0.1103 SP1) programs were used for instrument control, data acquisition and data analysis. The separations were carried out on a Hypersil gold C18 column (50 × 2.1 mm, 1.9 μm). The analysis was done under isocratic conditions at the flow rate of 0.200 mL/min. The mobile phase consisted of 0.2% formic acid in water (5%), methanol (85%) and acetonitrile (10%); the injection volume was 5 μL, column temperature 25 °C. The detection was done at 300, 350, 400 and 450 nm. The mass spectrometer was operated in the positive mode. HESI-source parameters: source voltage 4.5 kV, capillary voltage 36 V, tube lens voltage

110.00 V, capillary temperature 275 °C, sheath and auxiliary gas flow ($N_2$) 50 and 8 (arbitrary units), respectively. The MS spectra were acquired by full range acquisition covering 100–1000 m/z. For fragmentation study, a data dependent scan was performed by deploying the collision-induced dissociation (CID). The normalized collision energy of the CID cell was set at 25 eV.

The stability of two mono- (5,8 and 7,10) and two diendoperoxides (7,10,5',8' and 5,8,5',8') of β-carotene was monitored in organic solvents (methanol, acetonitrile and acetone) during 22 days of storage in the dark at 22 °C. The analysis were done immediately after preparing the solutions and then after 24, 96, 144, 192, 360 and 528 h. The optical density of the solutions was adjusted to 0.5 at the absorption maximum. The solutions were kept in tightly closed vials to avoid solvent evaporation and the injections to the LC-MS/MS apparatus were done directly from the vials using an autosampler (Dionex, UltiMate 300).

**Singlet oxygen detection.** The singlet oxygen luminescence was detected in the photon counting mode using a photomultiplier module H10330-45 (Hamamatsu Photonics K. K., Hamamatsu City, Japan), equipped with a 1100 nm cut-on filter and additional dichroic narrow-band filters NBP, selectable from the spectral range 1150–1355 nm (NDC Infrared Engineering Ltd, Bates Road, Maldon, Essex, UK). The samples were excited at 454 nm by laser pulses (1 kHz repetition rate) generated in an integrated nanosecond DSS Nd:YAG laser system equipped with a narrow bandwidth optical parametric oscillator (NT242-1k-SH/SFG; Ekspla, Vilnius, Lithuania). Typically, the pulse energy was in the range of several hundred μJ and the signal was recorded for 20 s. For determination of the action spectra the signal was collected for 1 min and normalized to laser power. The measurements were done in 1 cm thick fluorescence quartz cells. The pigments at the concentration of 0.8, 4 and 8 μM in acetone solution were saturated with Ar or $O_2$ by purging with pure gases for 30 min. The first order luminescence decay fitting using the Levenberg–Marquardt algorithm and further data analysis were done using a self-developed software.

**Electron paramagnetic resonance.** The spin trapping experiments were performed on a Bruker EMX AA 1579 EPR spectrometer (Bruker BioSpin, Germany) operating at 9 GHz, using DMPO as the spin trap. The following parameters were applied: microwave power 10.6 mW, modulation amplitude 0.5 G, scan width 80 G, and scan time 84 s. A flat quartz cell containing a solution of 35 μM βCar-5,8-EPO and 87 mM DMPO in DMSO was placed in the spectrometer resonance cavity and irradiated in situ with blue light (11.6 mW/cm²) obtained from a 300 W high pressure compact arc xenon lamp equipped with a water filter, a heat reflecting hot mirror, and a dichroic filter transmitting light in the 402–508 nm range. The time-dependent accumulation of the DMPO-OOH spin adducts' signal was carried out for 50 min. The calibration measurements for the estimation of $O_2^{\bullet-}$ yield were performed analogously in the dark, using 2.5 mM $KO_2$ solution in DMSO containing 87 mM DMPO.

For the EPR measurements in the lipid-like environment, the multilayer POPC vesicles were used. The liposome suspension containing βCar-5,8-EPO (35 μM) and DMPO (80 mM) was illuminated for 25 and 70 min with white light (30 mW/cm²) using the same light source (without the dichroic filter) and the measurement parameters as above.

**Computations.** The ab initio computations were carried out using Gaussian 16 Rev. B.01[88]. For the ground-state calculations, the B3LYP potential and the 6-31 G(d,p) base were used, in consistency with a previously optimized methodology[41,51], whereas the open-shell potential UB3LYP was applied in the computations on the biradical structures. All the conformations obtained in the computations reached their energetic minima. The singlet excited states of the all-trans β-carotene, βCar-5,8-EPO, and βCar-7,10-EPO were calculated using the approximations within TD-DFT.

## Data availability

The coordinates (3WU2) for PSII and (2BHW) for LHCII were obtained from PDB. All other data are available from the corresponding author upon request. Source data are provided with this paper.

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

## Acknowledgements
The authors thank Mrs. Jessica Catapano for her assistance in the EPR measurements. The work has been in part supported by a grant from the National Science Centre Poland (grant No. 2019/33/B/NZ1/02418 to L.F.) and by the Ministry of Education, Science and Technological Development of the Republic of Serbia (grant No. 451-03-9/2021-14/200133 to D.C.). The open-access publication of this article was funded by the Priority Research Area BioS under the program "Excellence Initiative–Research University" at the Jagiellonian University in Krakow.

## Author contributions
M.Z., H., and W.R. isolated pigments, synthesized the endoperoxides, and performed the experiments. D.C. performed the MS experiments. J.F. elaborated the synthetic method, its scale-up, and the separation methods. M.D. performed measurements of the lifetime and yield of singlet oxygen luminescence. A.W.-B. performed the EPR experiments. M.P. designed and carried out the computations. L.F and M.P. conceived the project, designed the experiments, analyzed the data, and wrote the manuscript. All authors analyzed the data, discussed the results, and contributed to the manuscript.

## Competing interests
The authors declare no competing interests.
