## [Peer Review File · Nature Communications]

nature portfolio

Peer Review File

Draft OnlyREVIEWER COMMENTS

Reviewer #1 (Remarks to the Author):

The subject is interesting, but the results/interpretation/discussion can be improved if the authors run further calculations. For instance, the experiments were carried out in solution, but they did not incorporate solvent effects in the calculations; it is known that these effects are very important in this kind of work, so it would be advisable to include solvent effects in the calculations. It would also be interesting to compare the computed and experimental excitation energies, computed oscillator strengths and intensity of absorption bands. With these results the authors can identify the electronic nature of the excited states, which would be very interesting to enhance the discussion. As the proposed mechanism involves internal conversion and intersystem crossing mechanisms, the author could try to compute the relevant energetic regions on the most important potential energy hypersurfaces, which would give further support to the proposed photocatalytic mechanism. After the authors have considered these suggestions and run the calculations, the manuscript can be reconsidered for publication.

Reviewer #2 (Remarks to the Author):

The study is aimed to investigate in detail the mechanisms of the formation and breakdown of β Car-EPOs, and the involvement of $^{1}O_2$ in these processes.

Research on anti- and pro-oxidant character of β Car has grown significantly due to the scientific and technological interest of this subject. The actual reaction mechanisms are still not completely resolved, although several essential steps have already been described. In this context, the work is adding new interesting information.

However, not all the conclusions drawn are fully demonstrated and convincing and, in my opinion, at this stage the work is more suitable for more specialized chemistry journals.

The following points should be addressed.

It is surprising to me that, in solution, β Car is able to fully protect BPheo from photodegradation (Fig2a), while becoming oxygenated up to 80% in 120 min. Apparently, when the [β Car] is reduced to

20% it is still able to fully protect the BPheo. Is β Car added in large excess? A comment on this is required and the concentration of the two pigments in solution must be given.

Another unexpected result shown in Fig.2a is that in 120 min the % degradation of BPheo in the absence of β Car is much less compared to that of β Car, meaning that the oxygenation of β Car is more severe of the oxidation of BPheo although β Car is known to have a very low intrinsic triplet yield that the authors claim to be the origin of the oxygenation of β Car. How do they explain that? In other words, more quantitative analysis has to be done.

The authors state that: "Not only is extended Ar purging insufficient to stop the reaction, it even runs under very low oxygen content ... Furthermore, very slow oxygenation of β Car also occurs in the dark, either with or without PhC, and white light or even red light above 630 nm accelerates it... In all these cases, the same pattern of β Car oxygenation is found (Fig. 2)".

Here all the statements require "quantitative" data on the product yields, since degradation of the β Car before use and ambient light exposure are possible reasons for the observed products (if present in small amount). Moreover, experiments with complete removal of oxygen by freeze and pumping are required to stop completely the reaction and rule out the presence of degradation products "before" measurements or photoexcitation effect on the solvent used.

The occurrence of a complex [3β Car- $3O_2$]* as responsible for the β Car-EPOs formation is only speculative, as well as the biradical nature of the β Car triplet state (no calculations are reported to give insight into this, while literature calculations on carotenoid triplet state, as far as I know, do not reveal such a nature). Different solvents, able to differently stabilize the complex should produce effects on the yield of products. I couldn't find data in the paper concerning the use of different solvents but experiments are required.

β Car triplet yield is very low in solution. Did the authors try to measure the triplet level in their experimental conditions? Can the very low triplet yield be responsible for the observed β Car-EPO yield in the absence of Bpheo?

The authors state that, in the breakdown of β Car-EPOs, the occurrence of biradical intermediates during O_2 release, seems to be consistent with a high amount of free radicals, since the trapped free radicals are higher than the amount of β Car-EPO used in the experiment by almost an order of magnitude. I do not see where the radicals come from. Which is the source of O-O bound to DMPO (Fig. 3d) if it is not coming from the β Car-EPOs? Please explain.

In terms of biological implications, I have some doubt. If correct, the proposed mechanisms would lead to a fast degradation of pigments in antenna complexes due to the oxidation of Car formed by transfer from Chl triplet states followed by production of singlet oxygen, both by Car-EPOs and by

Chl triplets which are not quenched anymore by closely bound Car after their oxidation. This is in contrast to the resistance of antenna complexes to light irradiation. Please comment on this.

Minor points:

Fig2a. Light fluxes conditions in photodegradation are not reported. Concentrations are also missing data.

Row 314 fig2a instead of fig3.a

Reviewer #3 (Remarks to the Author):

This paper is about a very important process in photosynthesis and in the biological control of reactive oxygen species:

the interaction between oxygen (in its triplet ground state as well in its first excited singlet state) with carotenes (and in general with carotenoids). The most abundant β -carotene is investigated here in detail with respect to the interaction with light, oxygen and singlet oxygen sensitizers that can also serve as triplet PeT transfer catalysts. Behind this investigation many questions appear that are discussed in the photobiological / photochemical community since decades and this contribution delivers a) remarkable experimental results and b) interesting interpretations with important biological / mechanistic consequences.

The photo-synthetic community will find numerous aspects that are worth to be exceedingly discussed with severe emphasis. From this point of view, this is fine paper that should be published.

In more detail, the authors discuss whether quenching of singlet oxygen that is a kind of undesired side product in photosynthesis is quenched by β -carotene in a dual fashion: by physical deactivation and formation of triplet β -carotene (where is no doubt also with the results described herein) as well as by chemical quenching and formation of endoperoxides. The latter process is under discussion since a long time, especially because of the unusual 7,10-endoperoxides that obviously point to non-concerted cycloaddition reactions. Additionally, the chemical efficiency of the formation of endoperoxide mixtures is very low and points to a dominating physical process and a minor pathway that can be initiated by radical-type or autoxidative reactions.

The authors have investigated the singlet oxygen reactions of β -carotene and from trapping / solvent deuteration studies, they conclude that these are not singlet oxygen reactions. At least the final endoperoxidation step is not a reaction of singlet oxygen with ground-state β -carotene - and from all data, I do agree with this interpretation (which is a truly important one).

The alternative process is a trapping reaction which indicates that oxygen is involved in its ground-state triplet form. Pure ground-state reactions (β -Car and oxygen) seem also to happen but are too slow to establish an explanation. Thus, only a combination of triplet β -Car and triplet oxygen explains the results. When I understand the interpretation correctly, the data does also not support the assumption that the major source of triplet Car is the physical quenching process of singlet oxygen by ground-state β -Car (because in this case also the concentration / DABCO / deuterated solvent - experiments are expected to show a significant influence). What is left over, is ISC of directly excited β -Car (very inefficient, but solvent independent) or an PeT from dyes to the singlet β -Car.

Another part of this work concerns the thermal and photochemical stability of the different endoperoxides formed from oxygen / β -Car reactions. The authors do not give any structural suggestions but show a bathochromic absorption shift for the 7,10, that is not present for the 5,8. And they mention the large increase in polarity of the products. I expect the endoperoxides under protic conditions to be reactive in the so-called Kornblum-De La Mare reactions which would lead to a) more polar hydroxy carbonyl products and b) bathochromic shifts because there appears are additional conjugating carbonyl group. For the 7,10-endoperoxide, the cyclic peroxide is more prone to ring-opening because of less steric hindrance and this would also explain the lower stability.

Methodology and Theory: I do not have any objections;

Reproducibility and data details: I do not have any objections.

More comments to more specific points:

- p6. when discussing the absorption effects from endoperoxide formation, please write "...., from 11 conjugated CC double bonds to 9,7, and eventually to 6 conjugated CC double bonds in....."

Essential is the word "conjugated" here and mentioning also the 9 and 7 systems help to understand the sequence of absorption spectra;

- is PhC a good abbreviation for photocatalyst? The photochemistry community uses PC, but PhotoCat would also be fine;

- in the Conclusion: triplet, not triple

- one should at the relevant place mention that the physical quenching process of singlet oxygen by ground-state β -Car can proceed with near diffusion limiting rate constant (please cite also the correct data directly in the paper) because of spin rules that allow 100% spin transfer. This process is in principle reversible - and for β -Car the energetics do not really favor one of the two processes - but because of spin rules the sensitization of singlet oxygen by triplet β -Car can occur only with 1/9 of diffusion. That might also contribute to the explanations of the authors;

- the trapping of triplet biradicals (and the β -Car is much more a triplet biradical than most triplet excited states because of the possible orthogonalization by C=C bond rotation) by ground-state triplet oxygen is an established method to detect these triplet biradical species, e.g. in Wirz et al., JACS 1993, 5400-5409 and many more papers from the Wirz, Scaiano, Adam groups. These papers should also be mentioned because they support the assumptions of the authors.

In summary, I strongly recommend publication of this paper (also exciting to read) because it has a sound set of data and will lead to a lot of discussion in the photobiology community and initiate more research in this important field.

Draft Only

Detailed response to the reviewers' comments

Reviewer #1 (Remarks to the Author):

The subject is interesting, but the results/interpretation/discussion can be improved if the authors run further calculations. For instance, the experiments were carried out in solution, but they did not incorporate solvent effects in the calculations; it is known that these effects are very important in this kind of work, so it would be advisable to include solvent effects in the calculations.

We are very grateful for reviewing our manuscript and for the insightful comments.

We agree that solvent effects are of key importance in most chemical systems, as well as in our system, and we investigated them both experimentally and computationally. In our computational approach, as developed earlier by us (see e.g. Fiedor et al., *Angew. Chem.* 2018), the influence of the solvent on the electronic properties of the pigments studied and on the molecular factors determining their stability and reactivity has been analyzed. As a matter of fact, no significant solvent effects have been found, and therefore, they were not shown. Also, for the limited space available (due to direct transfer from another journal), not all of the computational results could have been shown in the original manuscript. Indeed, the calculations show only negligible solvent effects on the electronic structure, stability, intramolecular steric strains, and conformations of the pigments, very consistent with the experimental data, and so they are not shown. Now, to provide the readers with evidence for the lack of such effects, we present a full set of the ab initio computed data in the Supplementary Information and refer to it in more detail in the revised text. For this purpose and to comply with the reviewer's remark, the Supplementary Table 2 was added in the revision and a new paragraph is added to the Results and Discussion section (p. 7).

It would also be interesting the compared the computed and experimental excitation energies, computed oscillator strengths and intensity of absorption bands.

As mentioned above, such computations were our starting point in this work, and their most relevant results were already included in part as the Supplementary Information (Suppl. Table 1) and discussed in the manuscript. They were very helpful in confirming the direction of the breakdown reactions of β Car-EPOs and helped us to confirm the identity of the hydrocarbon products of the O₂ release from β Car-EPOs. At the level of theory used in our computations, most properties of the ground states of the molecules (ground state energy, dipole moment, first- and higher-order polarizabilities, electron density distribution in HOMO and LUMO, the relative positions of the states, etc.) can be reliably predicted, whereas the excited state

properties, even the basic ones, are still beyond reach. Any further comparisons between the experimental and computational results would not be justified. For instance, the calculations (in general, not only ours) do not reproduce the energies of the singlet excited state in simple carotenoids: a ~100 nm difference in the S₂ band position (see Suppl. Table 1) is considered standard (in numerous works published) but excludes any fair comparison to the experiment.

With these results the authors can identify the electronic nature of the excited states, which would be very interesting to enhance the discussion. As the proposed mechanism involves internal conversion and intersystem crossing mechanisms, the author could try to compute the relevant energetic regions on the most important potential energies hypersurfaces, which would give further support to the proposed photocatalytic mechanism. After the authors have considered these suggestions and run the calculations, the manuscript can be reconsidered for publication.

As mentioned above, these goals are beyond the capabilities of ab initio approaches for Crts; which cannot properly deal with more basic issues such as predictions of the first excited triplet state and dark singlet states. Therefore, the mechanisms of IC and ISC in Crts cannot be adequately treated computationally. The same problems concern the computations on the intersections of the potential energy hypersurfaces and they remain beyond the reach of computational ab initio approaches. Another large difficulty stems from the fact that potential energies hypersurfaces of even a simple Crt each span over few hundreds (282 for βCar) normal modes of these molecules. With such limitations, our computational tools are only suitable for the treatment of radical species in their ground states and we have applied them in the analysis of stabilities and conformations of the biradical intermediates 1 and 2 which are key to the mechanism of βCar-EPO formation (Scheme 1A). These results (shown in Supplementary Table 1) have already been used in the discussion to support our conclusions.

Reviewer #2 (Remarks to the Author):

The study is aimed to investigate in detail the mechanisms of the formation and breakdown of βCar-EPOs, and the involvement of 1O₂ in these processes.

Research on anti- and pro-oxidant character of βCar has grown significantly due to the scientific and technological interest of this subject. The actual reaction mechanisms are still not completely resolved, although several essential steps have already been described. In this context, the work is adding new interesting information.

However, not all the conclusions drawn are fully demonstrated and convincing and, in my opinion, at this stage the work is more suitable for more specialized chemistry journals. The following points should be addressed.

We agree with this general impression, partly stemming from the fact that the manuscript has been transferred from a chemistry journal directly to Nature Comm. and we had no chance to elaborate its original version, which necessarily had to be compacted in order to stick to the

submission requirements. On the other hand, this is in some way in contrast to the opinion of the other reviewer, who states:

“this contribution delivers a) remarkable experimental results and b) interesting interpretations with important biological / mechanistic consequences”.

Surely, we are very grateful for reviewing our manuscript and for all the critical and insightful comments, and below we show how each comment was dealt with. We truly think that all of them were very to the point and helped us to substantially improve our manuscript.

It is surprising to me that, in solution, β Car is able to fully protect BPheo a from photodegradation (Fig2a), while becoming oxygenated up to 80% in 120 min. Apparently, when the [β Car] is reduced to 20% it is still able to fully protect the BPheo. Is β Car added in large excess? A comment on this is required and the concentration of the two pigments in solution must be given.

We admit, this crucial information concerning our preparative system was hard to find. It was not provided directly in the figure caption, while in the M&M section we only referred to our earlier work in which the synthetic method is described in full detail. In the revision, the relevant experimental details are given directly in the figure captions, not only to this particular figure.

[β Car] in this system is relatively high (50-70 μ M), which is perhaps not a large excess over [BPheo], but high enough for β Car to act efficiently as a photoprotector (physical quencher) while being consumed to a high degree in chemical reactions. However, one has to remember that the main products of these reactions, β Car-EPOs, are carotenoids themselves and as such they efficiently quench $^1\text{O}_2$, which is shown in our experiments (Suppl. Fig. 5) and discussed in the text. Moreover, the other products of β Car oxygenation, of lower molecular masses, such as cyclocitral, carotenals, etc., can act as quenchers too.

Another unexpected result shown in Fig.2a is that in 120 min the % degradation of BPheo in the absence of β Car is much less compared to that of β Car, meaning that the oxygenation of β Car is more severe of the oxidation of BPheo although β Car is known to have a very low intrinsic triplet yield that the authors claim to be the origin of the oxygenation of β Car. How do they explain that? In other words, more quantitative analysis has to be done.

We fully agree that “ β Car is known to have a very low intrinsic triplet yield” (due to an extremely low ISC efficiency in this molecule), but we nowhere claim that this triplet state is the origin of β Car oxygenation in our synthetic /preparative system. This seems to be a

misunderstanding. On the contrary, we claim that, to be efficient, these oxygenation reactions require a photocatalyst; in all parts of the figure mentioned by the reviewer, only the situations with BPheo as the photocatalyst are shown. The role of the photocatalyst in this system is to continuously supply triplet excitations. This is the point of our 'story'.

There is only a single instance where ISC in β Car may come into play. A very slow oxygenation of highly purified β Car occurs under red light and the oxygenation pattern closely resembles that seen in the presence of the photocatalyst. This suggests that the T1 state of β Car produced via direct photoexcitation (refs 55 and 56) is involved in this slow reaction. To stress this point, we now include an additional supplementary figure that shows the purity of the substrate β Car and the pattern of red-light-induced oxygenation of β Car (Suppl. Fig. 3).

The authors state that:” Not only is extended Ar purging insufficient to stop the reaction, it even runs under very low oxygen content ... Furthermore, very slow oxygenation of β Car also occurs in the dark, either with or without PhC, and white light or even red light above 630 nm accelerates it... In all these cases, the same pattern of β Car oxygenation is found (Fig. 2)”.

Here all the statements require “quantitative” data on the product yields, since degradation of the β Car before use and ambient light exposure are possible reasons for the observed products (if present in small amount). Moreover, experiments with complete removal of oxygen by freeze and pumping are required to stop completely the reaction and rule out the presence of degradation products "before" measurements or photoexcitation effect on the solvent used.

We are well aware of the chemical and photochemical instability of β Car. The possibility that “degradation of the β Car before use and exposure to ambient light are possible reasons for the observed products” in our experimental setup is entirely excluded from the very beginning. In all preparative runs, in which the products were analyzed and identified, only highly purified β Car has been used, with no trace impurities at all. However, this was not explicitly stated in the manuscript (for us, it was self-understood that we do not analyze degradation products). To stress it now, in the supplementary information we add a chromatogram showing the purity of our substrate (Suppl. Fig. 3) and the experimental conditions are explicitly mentioned in the text. Thus, the presence of degradation products prior to the reactions and solvent effects have been ruled out. To be on the safe side and to exclude any interference from acetone in the triplet-mediated processes (a known effect of carbonyls), for illumination of the samples we have used light in the red region of the spectrum (>630 nm, blue light was used only in the EPR experiment).

Sure enough, we have also considered experiments with complete removal of O₂ from the reaction medium. First, however, the only expected effect would be a halt of oxygenation, which is trivial. So, the complete removal of O₂ appeared to be in a way pointless and not very informative. Instead, we took another approach, as it made more sense to gain control over the content of ¹O₂ and its lifetime. However, technically speaking, the application of 'freeze and pump cycles' in our case is not quite feasible because for this single purpose we would have to completely redesign our experimental model and setup. For instance, our photochemical reactor cannot be placed under high vacuum, whereas our primary aim was to maintain the same conditions in all the synthetic runs, thus enabling a straightforward comparison of the products and a comparison to our earlier results (refs 17 and 18). Therefore, to decrease partial O₂ pressure we have purged it with high purity Ar (99.999). To further reduce partial pressure of O₂, after degassing the solvent under moderate vacuum, an extensive (3-4 days) chemical trapping (Oxoid™ AnaeroGen™ 2.5L sachets) was performed in our preparative system. The level of trace oxygen was monitored by recording the phosphorescence of Pd-Pheo, an O₂ indicator synthesized in our laboratory and used in our research (Kotkowiak et al. Angew. Chem. 2016). We now make this point more clear in the description of our experiments. Intriguingly, oxygenation of βCar at such low oxygen content does not come to a complete halt, most likely because some residual oxygen is occluded in noncovalent complexes with βCar.

The occurrence of a complex [βCar-3O₂]* as responsible for the βCar-EPOs formation is only speculative, as well as the biradical nature of the βCar triplet state (no calculations are reported to give insight into this, while literature calculations on carotenoid triplet state, as far as I know, do not reveal such a nature).

At this point, we have to disagree. The existence of [βCar-³O₂]* is not speculative at all, and strictly speaking, all species in their triplet states have biradical character, perhaps best exemplified in ³O₂ (ref Borden). Moreover, the presence of unpaired electrons in the triplet states of Crts (and other molecules) is best evidenced in their EPR spectra (refs 26 and 33). Yet, the above remarks indicate that our point was made not clear enough in the manuscript. The formation of collision complexes between ground state molecular oxygen and organic substrates was postulated by Schlenck already around 1950 (ref. 48) and the fact that βCar and even the short-lived ¹O₂ form [βCar-¹O₂]* is widely accepted (ref 32). What is more important, after the intracomplex energy transfer (i.e. the quenching), [βCar-¹O₂]* necessarily converts exactly into [βCar-³O₂]*, which then falls apart into βCar and ³O₂. The

cis-trans isomerization of Crts is another chemical reaction in which the biradical character of their T_1 state is strongly manifested.

To this end, for clarification, we now add an explanatory paragraph in the Results and Discussion section (p. 13).

Regarding the computations on Crt triplets, we agree that they do not reveal much but simply because the available quantum mechanical tools are too coarse. Even a very massive computational study of Hu et al. (ref. 62) in many aspects, including the nature of the dark S_1 state, is not conclusive. Unfortunately, more subtle effects are beyond the reach, so far.

Different solvents, able to differently stabilize the complex should produce effects on the yield of products. I couldn't find data in the paper concerning the use of different solvents but experiments are required.

The solvent effects would be very interesting to study, but it is easier said than done. The solvents were screened and solvent effects on the oxygenation were studied by us already some 20 years ago, when our interest turned to β Car-EPOs. The conclusion of this initial work was that they can be optimally synthesized in one handy solvent, namely acetone, a solvent of choice. And, after at least several years of using our system, we see that it is not just a matter of e.g. differences in encounter-complex stabilization by particular solvents. Rather, it is a question of delicate balance between the substrate, the photocatalyst, and the oxygen solubility, which cannot be easily achieved in other media. For instance, both β Car and BPheo are poorly soluble in n-hexane, β Car does not dissolve well in most alcohols, and BPheo in acetonitrile. Whereas, aprotic polar solvents, such as DMSO or DMF, apparently very suitable in this kind of synthetic work, cannot be conveniently and promptly separated from the reaction products, which excludes the reliable analyses of the latter. How possibly are the comparable reaction conditions to be maintained among various solvents for any quantitative analysis? Instead, when focusing on the role of singlet oxygen, we have applied a more elaborate approach, e.g. using deuterated acetone (“isotope trick”), and attempted to control the lifetime/content of singlet oxygen. Wherever possible, various solvents were used to study the degradation of β Car-EPOs.

By the way, the formation of β Car-EPOs occurs in a variety of media, starting from photosynthetic membranes in thylakoids (Chls as the source of triplets), through acetone (Chls

as the source of triplets, this work) and toluene/methanol mixture (rose Bengal or methylene blue as the source of triplets, (refs 16 and 22).

β Car triplet yield is very low in solution. Did the authors try to measure the triplet level in their experimental conditions? Can the very low triplet yield be responsible for the observed β Car-EPO yield in the absence of Bphea?

That is correct - the ISC quantum yield in β Car is extremely low ($\sim 0.1\%$), well below the detection thresholds for most spectroscopic techniques. This poses a major obstacle in obtaining any reliable direct characterization of this process in Crts (very few reports available in the literature). For instance, photolysis is used in the investigations of Crt triplets in photosynthetic pigment-protein complexes, but only those generated with high yields via T-T energy transfer from photosensitizers. Such an instrumentation is available to us (Drzewiecka-Matuszek et al., JBIC 2005) but the detection of intrinsically-generated β Car triplets in solution remains entirely out of question. Their “indirect” studies rely on a built-up approach, e.g. the detection of stable products generated from the triplets, such as carotenoid isomers. Here, we have used this approach to accumulate the products of very slow red light-induced oxygenation of β Car, using highly purified β Car as the substrate (virtually zero impurities background, Supplementary Fig. 3). Because the same oxygenation products (a series of β Car-EPOs) as in photocatalytic oxygenation were found, but with very low yield, we have attributed it to the reactivity of the intrinsically generated β Car triplet.

The authors state that, in the breakdown of β Car-EPOs, the occurrence of biradical intermediates during O₂ release, seems to be consistent with a high amount of free radicals, since the trapped free radicals are higher than the amount of β Car-EPO used in the experiment by almost an order of magnitude. I do not see where the radicals come from. Which is the source of O-O bound to DMPO (Fig. 3d) if it is not coming from the β Car-EPOs? Please explain.

We thank the reviewer for spotting this inconsistency. In Scheme 1b, of the many photochemically induced breakdown pathways that lead to the bleaching of EPOs, only the one responsible for the release of ¹O₂ is shown (confirmed by the detection of its luminescence during illumination of β Car-EPO under Ar). We do not claim that this is the main pathway; on the contrary, the release of ¹O₂ occurs with a relatively low efficiency (blue curve in Fig. 3a). This was not clearly stated in the manuscript; now a clarification is added under Scheme 1. Such a breakdown pathway becomes the main one only in the dark, since we observe a 1:1 conversion of β Car-EPOs to carotenes (Fig. 1b-1d) of m/z = 536 (mono-EPO m/z = 568), which implies O₂ release.

The photochemical breakdown of β Car-EPOs and their degradation products must be responsible for a high yield of free radicals. β Car-EPOs bleach under illumination, that is, undergo total degradation through multiple cleavage of the C-C bond, possibly, first in a Kornblum-De La Mare reaction (known for endoperoxides, as noted by reviewer#3) and then in light-promoted Norrish type II reactions, producing a variety of (carbon-centered) free radical products (= cascade); their consecutive reactions (propagation) with O_2 give multiple peroxy radicals (= cascade) which can be trapped by DMPO in our EPR experiments (in DMSO).

To verify the biological relevance of this pathway, we have carried out new EPR experiments in which the production of free radical species from β Car-EPOs was evaluated in the liposomes under illumination. Initially, the DMPO-OH adduct was detected as the product of a spontaneous decay of DMPO-OOH in an aqueous milieu. Upon prolonged illumination, a clear EPR spectrum of the DMPO-CH₃ adduct was recorded, as a result of the reaction of DMPO with secondary carbon-centered radicals, as generated in lipid degradation due to the cascade of (primary) free radicals triggered along β Car-EPO decay. This result supports the free radical mechanism of β Car-EPOs photodegradation and their strong pro-oxidant activity. The results of the new experiment are now shown in a new Suppl. Fig. 6 and a respective new paragraph is added in the revision (p. 20).

In terms of biological implications, I have some doubt. If correct, the proposed mechanisms would lead to a fast degradation of pigments in antenna complexes due to the oxidation of Car formed by transfer from Chl triplet states followed by production of singlet oxygen, both by Car-EPOs and by Chl triplets which are not quenched anymore by closely bound Car after their oxidation. This is in contrast to the resistance of antenna complexes to light irradiation. Please comment on this.

We appreciate this question because it is important, in light of our findings, to know how the photoprotective mechanisms present in the photosynthetic apparatus deal with triplet overproduction. It is not entirely true that LHCs are so resistant to excess light. As reported by Zolla et al. and Olszowka et al., they are also prone to photodegradation, but indeed, in vivo, the PSII core is much more vulnerable to photodamage caused by 1O_2 , especially to the D1/D2 proteins, i.e., the site where the triplets are produced (via charge recombination). Concerning the pigments, accumulation of β Car in the core complexes and xanthophylls in LHCs is a common feature of diverse photosynthetic organisms. To clarify this relevant issue, we add a new paragraph under “Implications for biological systems” (pp 28-29) and depict the role of the triplet state in photoprotective functioning of β Car in the photosystems in new

Scheme 4. The difference between β Car and xanthophylls in their binding to the photosynthetic proteins and their exposure to O₂ is shown in the new Supplementary Fig. 7.

Minor points:

Fig2a. Light fluxes conditions in photodegradation are not reported. Concentrations are also missing data.

The missing relevant information is now added

Row 314 fig2a instead of fig3.a

corrected

Reviewer #3 (Remarks to the Author):

Another part of this work concerns the thermal and photochemical stability of the different endoperoxides formed from oxygen / β -Car reactions. The authors do not give any structural suggestions but show a bathochromic absorption shift for the 7,10, that is not present for the 5,8. And they mention the large increase in polarity of the products. I expect the endoperoxides under protic conditions to be reactive in the so-called Kornblum-De La Mare reactions which would lead to a) more polar hydroxy carbonyl products and b) bathochromic shifts because there appears are additional conjugating carbonyl group. For the 7,10-endoperoxide, the cyclic peroxide is more prone to ring-opening because of less steric hindrance and this would also explain the lower stability.

We thank the reviewer for the effort to evaluate our manuscript and do appreciate all the comments.

We do not observe such a dark reaction to occur; the EPOs are fairly stable in protic MeOH in the dark, and in each case the only product detected is a carotene (m/z 536, isomer of β Car), which is much less polar than the parental EPO, as confirmed by HPLC and HPLC-MS analyses. Spectroscopically, the same product appears in other solvents (e.g. acetone and DMSO). The bathochromic shift in the case of β Car-7,10-EPO is due to the increase in the conjugation size in the carotene product. This is also apparent from our computations, which also confirm a lower stability of β Car-7,10-EPO due to steric effects (Suppl. Table 1).

Nevertheless, we are thankful for the remark because in vivo the Kornblum-De La Mare reaction, as the first step of β Car-7,10-EPO degradation followed by light induced Norrish reactions, may be responsible for the formation of cyclocitral and apo-10'-carotenal. Whereas, β Car-5,8-EPO may lead to apo-8'-crotanal. Such a sequence of events would also explain the

abundant generation of free radicals by β Car-EPOs under illumination. We now show this pathway in a new panel (Scheme 1C).

More comments to more specific points:

- p6. when discussing the absorption effects from endoperoxide formation, please write "...., from 11 conjugated CC double bonds to 9,7, and eventually to 6 conjugated CC double bonds in...."

Essential is the word "conjugated" here and mentioning also the 9 and 7 systems help to understand the sequence of absorption spectra;

corrected as suggested

- is PhC a good abbreviation for photocatalyst? The photochemistry community uses PC, but PhotoCat would also be fine;

corrected to PC, as suggested

- in the Conclusion: triplet, not triple

corrected

- one should at the relevant place mention that the physical quenching process of singlet oxygen by ground-state β -Car can proceed with near diffusion limiting rate constant (please cite also the correct data directly in the paper) because of spin rules that allow 100% spin transfer. This process is in principle reversible - and for β -Car the energetics do not really favor one of the two processes - but because of spin rules the sensitization of singlet oxygen by triplet β -Car can occur only with 1/9 of diffusion. That might also contribute to the explanations of the authors;

The high rate of singlet oxygen quenching by Crts has been mentioned in the manuscript with a reference to the source of the numbers (line 83), now reformulated accordingly. We are not quite sure how the spin rules can be adequately applied to the paramagnetic reactants in their collision complexes and thus we only refer to unfavorable spin statistics.

- the trapping of triplet biradicals (and the β -Car is much more a triplet biradical than most triplet excited states because of the possible orthogonalization by C=C bond rotation) by ground-state triplet oxygen is an established method to detect these triplet biradical species, e.g. in Wirz et al., JACS 1993, 5400-5409 and many more papers from the Wirz, Scaiano, Adam groups. These papers should also be mentioned because they support the assumptions of the authors.

We thank for the remark. The relevant articles are now referred to in the revision.

REVIEWERS' COMMENTS

Reviewer #1 (Remarks to the Author):

I agree with the revised version of the manuscript.

Reviewer #2 (Remarks to the Author):

The authors responded convincingly to the points raised. Therefore I recommend the publication in the revised form.

Reviewer #3 (Remarks to the Author):

I have carefully studied the revised version of the manuscript. I am satisfied with the comments and additions that the authors have given or added with respect to my questions and remarks.

There were much more advanced remarks and questions from the two other reviewers that seem to be much nearer to the field. My impression is, that the topic of the paper is really hot and that there are numerous points that require further discussions and many more experiments in the area of reasearch (not only by the authors alone - this would be impossible).

Thus, this paper is worth to be published now and will lead to interesting discussions and induce new research (this is what I expect from a scientific paper).